# DTX3L-mediated TIRR nuclear export and degradation regulates DNA repair pathway choice and PARP inhibitor sensitivity

Qi Ye[1,4], Jian Ma [1,4] ✉, Zixi Wang[1], Lei Li [1], Tianjie Liu [1], Bin Wang[1], Lizhe Zhu[1], Yuzeshi Lei[1], Shan Xu[1], Ke Wang[1], Yanlin Jian[1], Bohan Ma[1], Yizeng Fan[1], Jing Liu [1], Yang Gao[1], Haojie Huang [2,3] ✉ & Lei Li [1] ✉

53BP1 plays an important role in DNA double-strand break (DSB) repair and this activity is negatively regulated by its interaction with Tudor interacting repair regulator (TIRR). However, how the TIRR-53BP1 repair axis is regulated in response to DNA damage remains elusive. Here, we demonstrate that TIRR is translocated to the cytoplasm and degraded upon DNA damage. Ubiquitination of TIRR at lysine 187 by DTX3L is a critical process that regulates NHEJ pathway activity and PARP inhibitor sensitivity by facilitating XPO1-mediated TIRR nuclear export and degradation after DNA damage. We show that DTX3L is overexpressed in prostate cancers in patients and that decreased expression of TIRR due to DTX3L overexpression impairs the negative regulatory effect of TIRR on 53BP1, which consequently induces HR deficiency and chromosomal instability and sensitizes prostate cancer cells to poly (ADP-ribose) polymerase (PARP) inhibitors. Our work reveals a dual action of DTX3L on TIRR degradation and nuclear exportation and identifies DTX3L as an upstream regulator of the TIRR-53BP1 axis that governs DNA repair pathway choice and PARP inhibitor sensitivity. These findings suggest that TIRR ubiquitination and DTX3L overexpression could be viable biomarkers predicting PARP inhibitor sensitivity in cancers.

Double-strand breaks (DSBs) present a major threat to genome stability and have been linked to chromosomal translocations and cancer. When a DSB occurs in DNA, the choice between the repair mechanisms homologous recombination (HR) and nonhomologous end joining (NHEJ) depends on a complicated network of molecular interactions that activate either 53BP1 or BRCA1[1,2]. The activation and inhibition of 53BP1 function are pivotal in maintaining the balance of competing DSB repair pathways and are regulated by multiple mechanisms[3–5]. In response to DNA damage, 53BP1 is recruited to damaged chromatin by recognizing histone H2A ubiquitylated at Lys15 (H2AK15ub) and histone H4 dimethylated at Lys20 (H4K20me2) in the nucleosome core particle[6–9]. We and others also reported that posttranslational modifications of 53BP1 could affect its association with chromatin[5,10,11]. It has been recently reported that Tudor-interacting repair regulator (TIRR), a paralog of the Nudix protein NUDT16, binds to the Tudor domains of 53BP1, which blocks the surface of 53BP1 to which H4K20me2 binds[12,13]. A variety of experimental studies have confirmed the crucial role of the TIRR protein as a natural switch between the NHEJ and HR pathways by negative regulation of 53BP1[12–16]. Although 53BP1 forms a tight complex with TIRR, the TIRR–53BP1 complex is

[1]Department of Urology, The First Affiliated Hospital of Xi'an Jiaotong University, Xi'an, China. [2]Department of Urology, The First Affiliated Hospital, Zhejiang University School of Medicine, Hangzhou, China. [3]Institute of Urologic Science and Technology, The First Affiliated Hospital, Zhejiang University School of Medicine, Hangzhou, China. [4]These authors contributed equally: Qi Ye, Jian Ma. ✉e-mail: majian0922@gmail.com; huanghaojie@zju.edu.cn; lilydr@163.com

dissociated in response to DNA damage, which is possibly triggered by RNA molecules and RAP1-interacting factor 1 (RIF1)[12–14]. However, how the TIRR protein responds to DNA damage and the destination of TIRR after dissociation from 53BP1 remain unknown.

Deltex-3-like (DTX3L), also known as B-lymphoma and BAL-associated protein (BBAP), is a ubiquitin E3 ligase belonging to the Deltex family[17,18]. Previous studies have reported that DTX3L forms a complex with PARP9 and ubiquitinates histones in certain contexts; for example, it monoubiquitylates histone H4 in response to DNA damage[19,20]. It also acts on host histone H2BJ to modify chromatin accessibility at interferon-stimulated genes[21]. Although it is known that DTX3L is involved in the response to DNA damage, the detailed function of DTX3L in DNA repair pathways remains unclear.

In the present study, we show that TIRR is translocated to the cytoplasm upon DNA damage. After dissociation from 53BP1 at DSBs, TIRR is ubiquitinated by DTX3L at lysine-187, which is near the nuclear export signal (NES) of TIRR, to facilitate the binding of XPO1 to TIRR and consequently lead to TIRR nuclear export and degradation in the cytoplasm. In prostate cancer cells with DTX3L overexpression, the negative regulatory effect of TIRR on 53BP1 is impaired, which induces HR deficiency and chromosomal instability and results in high sensitivity to poly (ADP-ribose) polymerase (PARP) inhibitors. Our work reveals DTX3L as a key regulator of TIRR function and DNA repair, suggesting that increased expression of DTX3L could be a target for PARP inhibitors in the treatment of cancer.

## Results

### TIRR is translocated to the cytoplasm after DNA damage
TIRR and 53BP1 interaction has been well characterized[12,13,16]. However, how the function of TIRR is regulated in response to DNA damage remains elusive. To this end, we employed different culture conditions and drugs, including IR, cisplatin, and CPT, to induce DNA damage in U2OS cells, which are commonly used for the analysis of DNA damage-induced protein focus formation (Fig. 1a). We found that TIRR accumulated in the nucleus before DNA damage but translocated to the cytoplasm after DNA damage was induced (Fig. 1a, b). Similar results were obtained in PC-3 prostate cancer cells (Supplementary Fig. 1a, b). To confirm this phenomenon, we transfected U2OS cells with Flag-tagged TIRR and subjected cells to IR or treated them with cisplatin or CPT (Fig. 1c). Consistently, almost all the Flag-tagged proteins were located in the cytoplasm, while 53BP1 formed foci after IR (Fig. 1c, d). We further examined the subcellular distribution of TIRR by cellular fractionation and western blotting in both U2OS and PC-3 cells. As expected, both endogenous and transfected TIRR accumulated in the cytoplasm after IR (Fig. 1e and Supplementary Fig. 1c, d). Moreover, endogenous TIRR in both U2OS and PC-3 cells was mostly located in the cytoplasm and did not overlap with 53BP1 foci in the nucleus after IR (Fig. 1f, g and Supplementary Fig. 1e, f). These data indicate that TIRR is translocated to the cytoplasm after dissociating from 53BP1 upon DNA damage.

### XPO1 mediates TIRR nuclear export in response to DNA damage
The above observations imply that TIRR shuttles from the nucleus to the cytoplasm upon DNA damage. To search for the nuclear exportation signal (NES) in TIRR protein, we performed in silico analysis and identified a consensus NES sequence φ-X (2–3)-φ-X (2–3)-φ-X-φ (where φ is Lys, Val, Ile, Phe or Met; X is any amino acid; and the numbers in parentheses denote the number of repeats), which can be recognized by the nuclear exportation protein XPO1[22–27] (Fig. 2a). Next, we sought to experimentally determine whether XPO1 is involved in the transportation of TIRR in response to DNA damage. Coimmunoprecipitation (co-IP) assays confirmed that endogenous XPO1 interacted with TIRR in PC-3 prostate cancer cells (Supplementary Fig. 2a). To determine which region of the TIRR NES mediates its interaction with XPO1, we transfected 293T cells with a series of plasmids

expressing Flag-tagged wild-type (WT) TIRR or Flag-tagged TIRR with various mutations in the NES and HA-tagged XPO1. Co-IP demonstrated that the M3 region of the NES (residues 173 to 182) was needed for TIRR to be recognized by XPO1 (Fig. 2a, b). Structural studies have demonstrated that I521, L525, F561, and F572 in XPO1 as key residues forming a hydrophobic groove essential for NES recognition[22,28]. Consequently, we performed a Co-IP assay with WT and mutant (I521A/L525A/F561A/F572A) XPO1 and found that these mutations abolished XPO1's binding to TIRR (Supplementary Fig. 2b). To investigate this in vitro, we synthesized the NES peptide and using Isothermal Titration Calorimetry (ITC) assay further confirmed the binding between NES (residues 173 to 182) and XPO1 (Supplementary Fig. 2c). Moreover, the interaction between TIRR and XPO1 was strengthened with higher IR doses and progressively increased over time following IR exposure (Supplementary Fig. 2d, e), indicating the XPO1/TIRR interaction is both dose-dependent and time-dependent in response to DNA damage. Furthermore, while WT TIRR and most of the TIRR NES mutants translocated to the cytoplasm, only the TIRR M3 NES mutant was retained in the nucleus after IR (Fig. 2c, d and Supplementary Fig. 2f). To confirm that XPO1 mediates TIRR translocation, we examined the subcellular distribution of TIRR in XPO1 knockout cells. Western blotting showed that XPO1 knockout significantly decreased cytoplasmic TIRR accumulation in U2OS cells after IR (Fig. 2e). Consistently, both XPO1 knockout and treatment with the XPO1 inhibitor KPT-330 abolished TIRR nuclear export in PC-3 cells after IR (Fig. 2f–i and Supplementary Fig. 2g). These data indicate that XPO1 recognizes the NES motif of TIRR and promotes TIRR nuclear export in response to DNA damage.

### TIRR is ubiquitinated and degraded after translocation to the cytoplasm
We noticed from our IFC data that the TIRR signal was progressively decreased over time following IR exposure (Fig. 3a–d). In agreement with this observation, we demonstrated that TIRR expression was downregulated at the protein level but not at the mRNA level after IR (Fig. 3e, f). We further observed increased ubiquitination of TIRR after DNA damage, indicating that TIRR undergoes proteasomal degradation in response to DNA damage (Fig. 3g). We further explored the influence of TIRR ubiquitination on XPO1-mediated TIRR translocation. Indeed, the TIRR-XPO1 interaction was markedly increased after induction of DNA damage by IR (Fig. 3h). To determine whether ubiquitination occurs before TIRR nuclear export, we used the E1 inhibitor MLN4924 to block TIRR ubiquitination and the proteasome inhibitor MG132 to prevent TIRR degradation. Intriguingly, MLN4924 but not MG132 blocked the TIRR-XPO1 interaction in both PC-3 and U2OS cells (Fig. 3i), indicating the TIRR ubiquitination is necessary for its nuclear export. Indeed, XPO1 knockout prevented TIRR degradation following DNA damage (Fig. 3j). Consistently, the nuclear export of TIRR was also abolished by MLN4924 but not MG132 in PC-3 and U2OS cells subjected to IR (Fig. 3k–n). These data suggest that ubiquitination is necessary for TIRR nuclear export and that TIRR is ubiquitinated before being transported to and degraded in the cytoplasm.

### DTX3L-mediated TIRR ubiquitination triggers TIRR nuclear translocation and degradation after DNA damage
To identify the potential E3 ubiquitin ligase that triggers TIRR ubiquitination in response to DNA damage, we performed LC–MS analysis to identify proteins that interact with TIRR in cells subjected to IR and control cells (Fig. 4a). We noticed that several E3 ubiquitin ligases showed increased binding to TIRR after IR, including TRIM26, TRIM56, DTX3L, and TRIP12 (Fig. 4b, c). We then sought to examine whether these E3 ubiquitin ligases regulate TIRR protein levels after DNA damage. When ectopically expressed, all the E3 ubiquitin ligases bound TIRR (Fig. 4d and Supplementary Fig. 3a–c); however, only

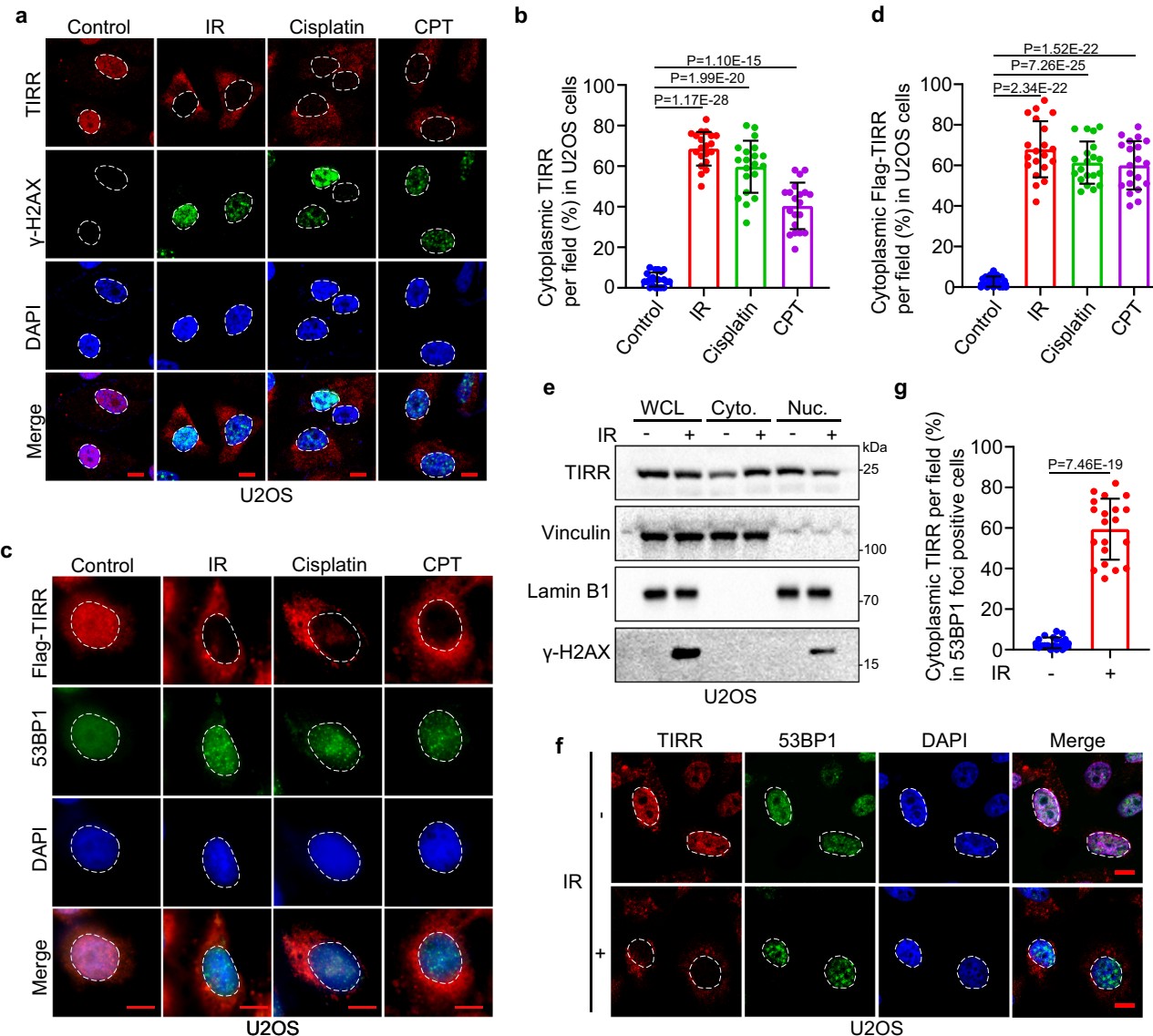

**Fig. 1 | TIRR is translocated to the cytoplasm after DNA damage.**
**a** Immunofluorescent cytochemistry (IFC) of TIRR and γ-H2AX in U2OS cells treated with IR (8 Gy), Cisplatin (10 μM, 24 h) or CPT (2 μM, 3 h). Scale bar, 10 μm. **b** Quantification of cells with cytoplasmic translocation of endogenous TIRR as shown in (**a**). Data were shown as the mean ± SD from 20 fields (>200 cells, $n = 20$) from three biological replicates. Two-tailed unpaired Student's $t$-test; $P$ values based on the order of appearance: 1.17E-28, 1.99E-20, 1.10E-15. **c** IFC of Flag-tagged TIRR and 53BP1 in U2OS cells transfected with Flag-TIRR and subjected to IR (8 Gy), Cisplatin (10 μM, 24 h) or CPT (2 μM, 3 h). Scale bar, 10 μm. **d** Quantification of U2OS cells with cytoplasmic translocation of Flag-TIRR, as shown in (**c**). Data were

shown as the mean ± SD from 20 fields (>200 cells, $n = 20$) from three biological replicates. Two-tailed unpaired Student's $t$-test; $P$ values based on the order of appearance: 2.34E-22, 7.26E-25, 1.52E-22. **e** WB analysis of whole-cell (WCL), cyto-solic (Cyto.), and nuclear (Nuc.) fractions from U2OS cells with or without IR (8 Gy). Similar results were obtained in three independent experiments. **f** IFC of TIRR and 53BP1 in U2OS cells treated with or without IR (8 Gy). Scale bar, 10 μm. **g** Quantification of cells with 53BP1 foci and cytoplasmic translocation of endo-genous TIRR as shown in (**f**). Data were shown as the mean ± SD from 20 fields (>200 cells, $n = 20$) from three biological replicates. Two-tailed unpaired Student's $t$-test; $P = 7.46$E-19. Source data are provided as a Source Data file.

ectopic expression of DTX3L decreased TIRR protein levels after DNA damage (Fig. 4e and Supplementary Fig. 3d–f). The decrease in both endogenous and exogenous TIRR protein expression after DNA damage was rescued by the proteasome inhibitor MG132 (Fig. 4e). Co-IP further confirmed the endogenous binding between TIRR and DTX3L (Fig. 4f, g). Consistent with the XPO1-TIRR interaction, the interaction between TIRR and DTX3L was also enhanced with increasing IR doses and progressively intensified over time following IR exposure (Supplementary Fig. 3g, h). Moreover, DTX3L expression augmented TIRR polyubiquitination in response to DNA damage, particularly after treatment with MG132, but not with MLN4924 (Fig. 4h and Supplementary Fig. 3i). On the other hand, DTX3L knockout attenuated TIRR polyubiquitination in PC-3 cells (Fig. 4i and

Supplementary Fig. 3j). Both the RING and DTC domains of DTX3L are reported to play essential roles in its ubiquitylation activity[21,29]. To investigate this, we constructed DTX3L truncations with deletions of the RING domain (D1), the DTC domain (D2), and both domains (D3) (Supplementary Fig. 3k). All three deletions resulted in reduced ubi-quitylation levels of TIRR (Supplementary Fig. 3l). Consistently, the catalytically inactive mutant of DTX3L[21] also abolished its ubiquitina-tion function on TIRR (Supplementary Fig. 3m). Thus, we revealed that DTX3L expression clearly shortened the TIRR protein half-life (Sup-plementary Fig. 3n, o). In contrast, DTX3L knockout prolonged the half-life of the endogenous TIRR protein in PC-3 cells (Fig. 4j, k). These results suggest that DTX3L is a bona fide E3 ligase that modulates TIRR protein ubiquitination and thus decreases TIRR stability in response to

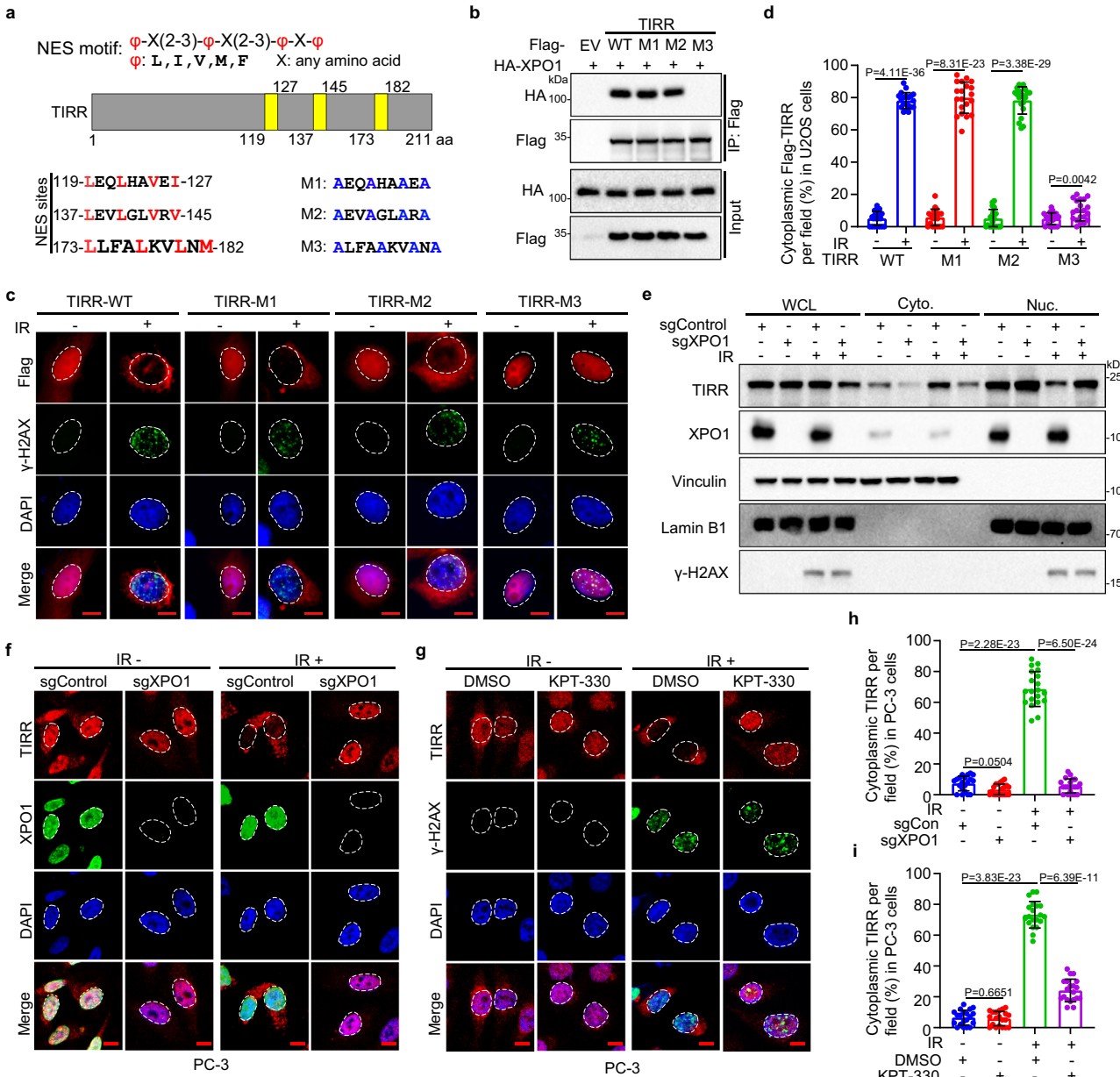

**Fig. 2 | XPO1 mediates TIRR nuclear export in response to DNA damage.**
**a** Schematic representation highlighting potential nuclear export signals (yellow) in TIRR and amino acid sequence alignments of NES motifs and TIRR NES mutants. aa, amino acid. **b** Co-IP of whole-cell lysates from 293T cells transfected with HA-XPO1, Flag-TIRR-WT, or Flag-TIRR NES mutants, and treated with IR (8 Gy). IFC (**c**) and quantification (**d**) of cytoplasmic translocation of Flag-TIRR-WT and NES mutants in U2OS cells transfected with indicated constructs and treated with or without IR (8 Gy). Scale bar, 10 μm. Data were shown as the mean ± SD from 20 fields (>200 cells, *n* = 20) from three biological replicates. Two-tailed unpaired Student's *t*-test; *P* values based on the order of appearance: 4.11E-36, 8.31E-23, 3.38E-29, 0.0042. **e** WB analysis of whole-cell (WCL), cytosolic (Cyto.) and nuclear (Nuc.) fractions

from control or XPO1 knockout PC-3 cells post-IR (8 Gy). IFC (**f**) and quantification (**h**) of cytoplasmic TIRR in control and XPO1 knockout PC-3 cells post-IR (8 Gy). Scale bar, 10 μm. Data were shown as the mean ± SD from 20 fields (>200 cells, *n* = 20) from three biological replicates. Two-tailed unpaired Student's *t*-test; *P* values based on the order of appearance: 0.0504, 2.28E-23, 6.50E-24. IFC (**g**) and quantification (**i**) of cytoplasmic translocation of TIRR in PC-3 cells pretreated with of DMSO or KPT-330 (1 μM, 12 h) post-IR (8 Gy). Scale bar, 10 μm. Data were shown as the mean ± SD from 20 fields (>200 cells, *n* = 20) from three biological replicates. Two-tailed unpaired Student's *t*-test; *P* values based on the order of appearance: 0.6651, 3.83E-23, 6.39E-11. Source data are provided as a Source Data file. Similar results for (**b**, **e**) panels were obtained in three independent experiments.

DNA damage. We next examined whether DTX3L expression affects TIRR binding with XPO1. While overexpression of DTX3L increased TIRR-XPO1 binding (Supplementary Fig. 3p), knockout of DTX3L abolished the effect of DNA damage in increasing the TIRR-XPO1 interaction (Fig. 4l). Consistently, DTX3L knockout increased the nuclear distribution of TIRR and abolished TIRR nuclear export after DNA damage (Fig. 4m, n and Supplementary Fig. 3q). Our data show that DTX3L-mediated ubiquitination of TIRR increases the interaction

of TIRR with XPO1 and promotes TIRR nuclear export after DNA damage.

### Ubiquitination of TIRR at lysine-187 by DTX3L dictates NHEJ pathway activity and PARP inhibitor sensitivity

To further understand how ubiquitination of TIRR influences its interaction with XPO1, we examined which lysine residues of TIRR are ubiquitinated by DTX3L (Fig. 5a). Intriguingly, mutagenesis analysis

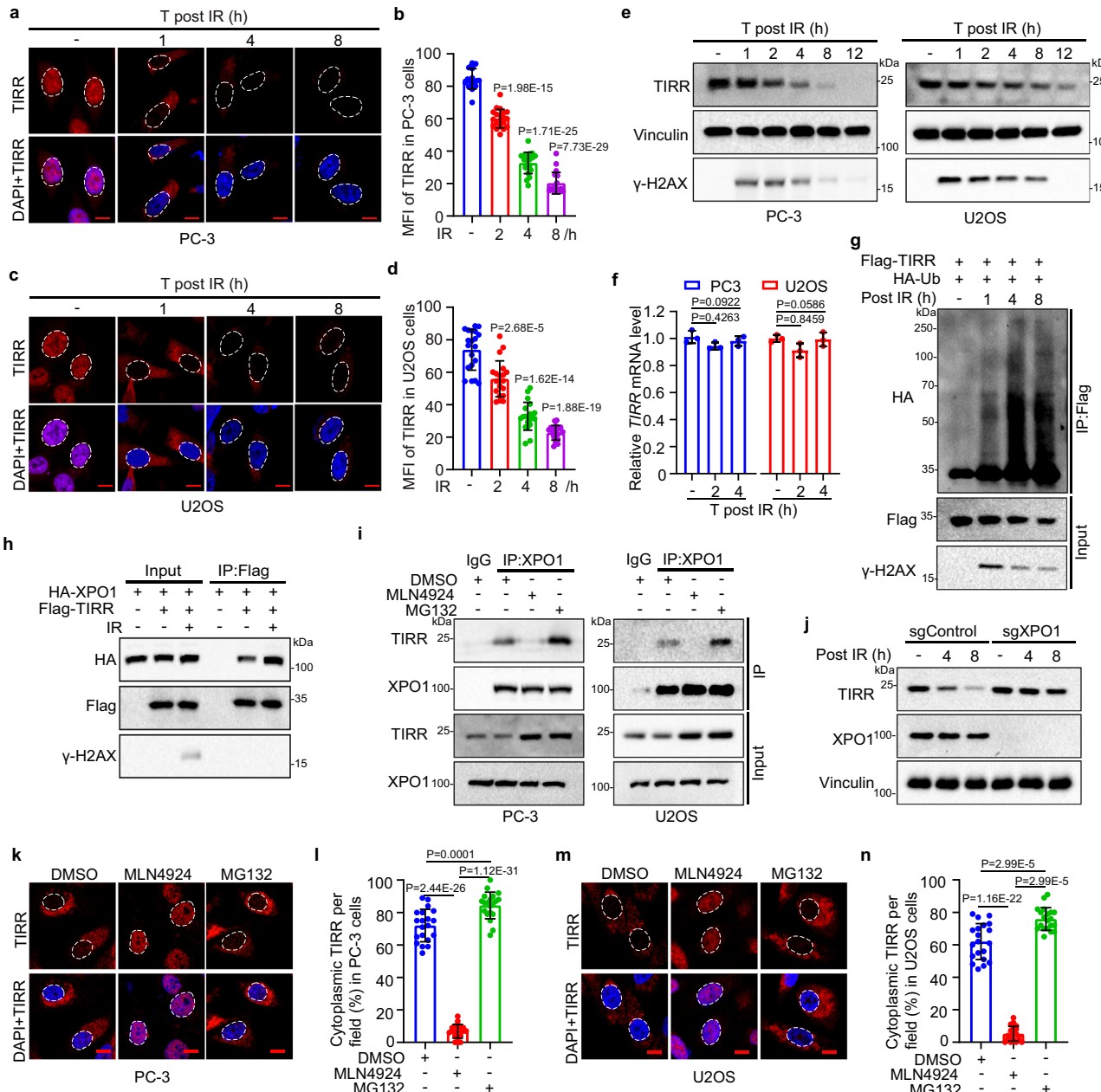

**Fig. 3 | TIRR is ubiquitinated and degraded after translocated to cytoplasm.** IFC illustrating TIRR signal and quantification of mean fluorescence intensity (MFI) in PC-3 (**a**, **b**) and U2OS (**c**, **d**) cells at indicated time points after IR (8 Gy) treatment. Scale bar, 10 μm. Data were shown as the mean ± SD of 20 fields (>200 cells, *n* = 20) from three biological replicates. Two-tailed unpaired Student's *t*-test; *P* values based on the order of appearance: 1.98E-15, 1.71E-25, 7.73E-29 (**b**); 2.68E-5, 1.62E-14, 1.88E-19 (**d**). **e** WB analysis of TIRR in whole-cell lysates of PC-3 and U2OS cells post-IR (8 Gy). **f** mRNA expression of *TIRR* in PC-3 or U2OS post-IR (8 Gy). Data were shown as the mean ± SD of three independent experiments (*n* = 3). Two-tailed unpaired Student's *t*-test; *P* values based on the order of appearance: 0.4263, 0.0922, 0.8459, 0.0586. **g** In vivo polyubiquitination assay in whole-cell lysates of 293T cells transfected with Flag-TIRR and HA-Ub, and harvested following IR (8 Gy)

at indicated time points. **h** Co-IP of whole-cell lysates from 293T cells transfected with HA-XPO1 and Flag-TIRR, followed by IR (4 Gy). **i** Co-IP of endogenous TIRR and XPO1 in whole-cell lysates of PC-3 and U2OS cells pretreated with DMSO, MLN4924, or MG132, and exposed to IR (8 Gy). **j** WB analysis of TIRR in whole-cell lysates from control or XPO1 knockout PC-3 cells post-IR (8 Gy). IFC and quantification of cytoplasmic TIRR in PC-3 cells (**k**, **l**) or U2OS cells (**m**, **n**) pretreated with DMSO, MLN4924, or MG132, followed by IR (8 Gy). Scale bar, 10 μm. Data were shown as the mean ± SD from 20 fields (>200 cells, *n* = 20) from three biological replicates. Two-tailed unpaired Student's *t*-test; *P* values based on the order of appearance: 2.44E-26, 0.0001, 1.12E-31 (**l**); 1.16E-22, 2.99E-5, 2.99E-5 (**n**). Source data are provided as a Source Data file. Similar results for (**e**, **g**, **h**, **i**, **j**) panels were obtained in three independent experiments.

showed that mutation of lysine-187 but not the other lysine residues in TIRR diminished DTX3L-mediated TIRR ubiquitination (Fig. 5b). Indeed, the ubiquitination at lysine-187 in TIRR was detected by mass spectrometry (Supplementary Fig. 4a). Moreover, mutation of lysine-187 to arginine or alanine abolished the DTX3L-mediated TIRR degradation and prolonged the TIRR protein half-life in PC-3 cells (Fig. 5c, d

and Supplementary Fig. 4b–d), suggesting that DTX3L promotes polyubiquitination of TIRR at lysine-187. Since lysine-187 is close to the NES of TIRR, we examined whether lysine-187 ubiquitination affects NES recognition by XPO1. Indeed, mutation of lysine-187 substantially inhibited the IR-induced increases in TIRR binding to XPO1 and the nuclear localization of TIRR (Fig. 5e–g and Supplementary Fig. 4e).

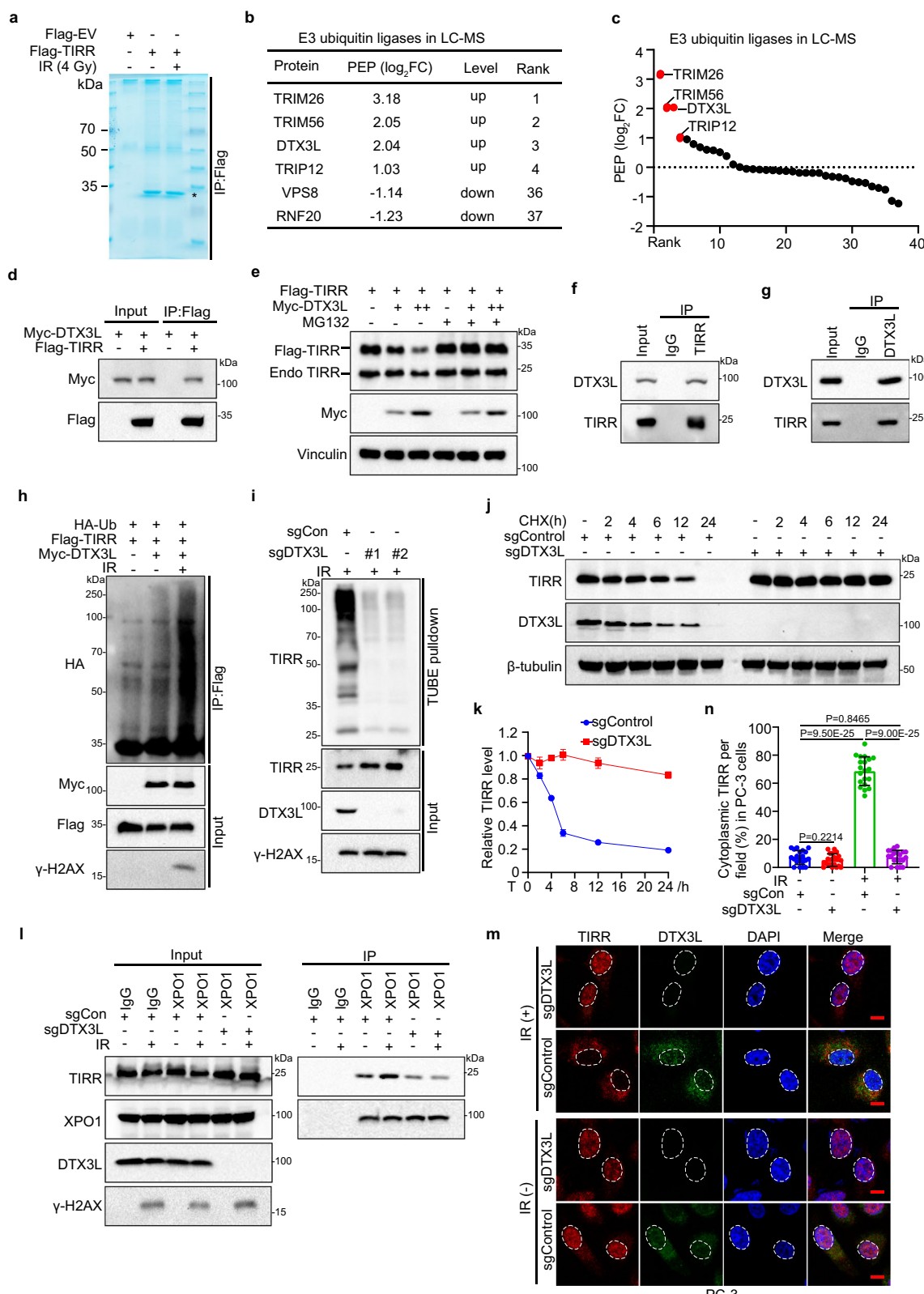

Given that TIRR acts as a natural switch between the NHEJ and HR pathways by negatively regulating 53BP1[12–16], we hypothesize that TIRR ubiquitination plays a crucial role in this process. Intriguingly, we found that the reduction in binding between TIRR and 53BP1 following DNA damage was abolished by lysine-187 mutation (Supplementary Fig. 4f). Furthermore, we observed that 53BP1 foci formation was nearly mutually exclusive with TIRR nuclear export upon DNA damage

and knockout of TIRR resulted in increased 53BP1 foci formation after IR treatment (Fig. 5h–j). Importantly, reconstitution of TIRR knockout cells with the TIRR K187R mutant markedly enhanced the inhibitory effect of TIRR on 53BP1 accumulation at DSB sites compared to WT TIRR or other KR mutants, suggesting that the retention of TIRR in the nucleus caused by lysine-187 mutation prolonged the inhibitory effect of the protein on 53BP1 after DNA damage (Fig. 5i, j). Consistent with

**Fig. 4 | DTX3L-mediated TIRR ubiquitination triggers TIRR nuclear translocation and degradation after DNA damage. a–c** Co-IP of whole-cell lysates from 293T cells transfected with Flag-TIRR were treated with or without IR (4 Gy) (**a**). The E3 ubiquitin ligases identified by mass spectrometry were listed with their PEP score fold changes (**b**) and plotted with the rank according to their PEP scores fold changes (**c**). **d** Co-IP of whole-cell lysates from 293T cells transfected with Myc-DTX3L and Flag-TIRR. **e** WB analysis of TIRR in whole-cell lysates from 293T cells transfected with Myc-DTX3L or Flag-TIRR, with or without pretreatment with MG132, 2 h post-IR (8 Gy). **f, g** Co-IP of endogenous TIRR and DTX3L in whole-cell lysates of PC-3 cells. **h** In vivo polyubiquitination assay of whole-cell lysates from 293T cells transfected with Flag-TIRR, Myc-DTX3L and HA-Ub, followed by IR (4 Gy). **i** TUBE pulldown assays in whole-cell lysates from control and DTX3L knockout PC-3 cells, pretreated with MG132 and exposed to IR (4 Gy). WB analysis of TIRR in whole-cell lysates at indicated time points of CHX treatment (**j**) and quantification (**k**) in control and DTX3L knockout PC-3 cells. Data were shown as the mean ± SD from three biological replicates (*n* = 3). **l** Co-IP between endogenous TIRR and XPO1 in whole-cell lysates from control or DTX3L knockout PC-3 cells, treated with or without IR (8 Gy). IFC (**m**) and quantification (**n**) of cytoplasmic translocation of TIRR in control and DTX3L knockout PC-3 cells with or without IR (8 Gy). Scale bar, 10 μm. Data were shown as the mean ± SD from 20 fields (>200 cells, *n* = 20) from three biological replicates. Two-tailed unpaired Student's *t*-test; *P* values based on the order of appearance: 0.2214, 9.50E-25, 0.8465, 9.00E-25. Source data are provided as a Source Data file. Similar results for (**d–j, l**) panels were obtained in three independent experiments.

these results, while TIRR knockout increased NHEJ reporter activity and decreased HR reporter activity, rescue expression of K187R mutant abolished this effect more substantially than WT TIRR (Fig. 5k, l). Furthermore, a dose-response study in U2OS cells showed that the IC50 of the PARP inhibitor olaparib was significantly lower in TIRR knockout cells compared to EV control cells. However, this effect was reversed by TIRR expression, particularly with the K187R mutant (Supplementary Fig. 4g). By performing colony formation assays, we confirmed that olaparib markedly inhibited growth of TIRR knockout cells with formation of fewer and smaller colonies, but only minimal effect was observed in EV control and TIRR WT/K187R-expressing cells (Supplementary Fig. 4h, i). Overall, our data indicate that ubiquitination of TIRR at lysine-187 is a critical process that regulates NHEJ pathway activity and PARP inhibitor sensitivity by facilitating TIRR nuclear export upon DNA damage.

## DTX3L overexpression impairs HR activity and promotes chromosomal instability in prostate cancer

Since TIRR ubiquitination is essential for DNA repair pathway choice and PARP inhibitor sensitivity, we aimed to identify clinical correlates of TIRR ubiquitination that could guide the use of PARP inhibitors. We noticed that both *DTX3L* and *TIRR* mRNA expression levels were increased in tumor tissues compared to normal tissues from The Cancer Genome Atlas (TCGA) prostate cancer datasets, and this increase was associated with the progression of prostate cancer (Supplementary Fig. 5a). We further examined TIRR and DTX3L protein expression in specimens from a cohort of prostate cancer patient (*n* = 44) using immunohistochemistry (IHC). We found that the protein levels of TIRR and DTX3L in patient specimens were negatively correlated (Fig. 6a, b). To mimic the pathophysiological features of prostate cancer in patients, we stably expressed DTX3L in PC-3 and U2OS cells (Supplementary Fig. 5b). Consistent with our previous finding, the DTX3L decreased TIRR protein levels, especially after DNA damage (Supplementary Fig. 5b). Moreover, we showed that DTX3L expression increased 53BP1 focus formation but inhibited BRCA1 focus formation at DSBs without influencing their protein levels in both PC-3 and U2OS cells (Fig. 6c–f and Supplementary Fig. 5c–f). Consequently, the expression of DTX3L decreased HR reporter activity but increased NHEJ reporter activity in PC-3 cells (Fig. 6g, h). Furthermore, camptothecin (CPT) treatment resulted in a marked increase in the number of chromosome breaks per cell in DTX3L-expressing PC-3 cells, aligning with the elevated frequency of asymmetric radial chromosome structures typically seen in HR-defective cells[4,10,11,30] (Fig. 6i, j). Consistently, analysis of The Cancer Genome Atlas (TCGA) prostate dataset showed that high DTX3L expression associated with the increased genomic instability signature[31–33], including weighted genomic integrity index (wGII) and HR deficiency score (HRD) (Fig. 6k, l). These data indicate that DTX3L was overexpressed in the prostate cancer patient specimens we examined and that DTX3L overexpression inhibits HR and induces chromosomal instability by downregulating TIRR after DNA damage.

## DTX3L expression sensitizes prostate cancer cells to synthetic lethality by PARP inhibitors

The fact that DTX3L was overexpressed in prostate cancer patient samples and that DTX3L overexpression inhibited HR and caused chromosomal instability prompted us to explore whether DTX3L overexpression sensitizes prostate cancer cells to synthetic lethality by PARP inhibitors. A dose-response study in both PC-3 and U2Os cells revealed that the IC50 of the PARP inhibitor olaparib in DTX3L-expressing cells was much lower than that in EV control cells (Fig. 7a, b). By performing colony formation assays, we confirmed that olaparib markedly inhibited the growth of DTX3L-expressing cells, allowing the formation of fewer and smaller colonies; however, olaparib had a minimal effect on EV control cells (Fig. 7c–f). Moreover, while DTX3L overexpression increased cell sensitivity to PARP inhibitors, the TIRR-K187R mutant reduced this sensitivity, indicating that the effect of DTX3L on cell sensitivity to PARP inhibitors depends on its ability to ubiquitinate TIRR (Supplementary Fig. 6). We further investigated the effect of olaparib in vivo. PC-3 xenograft tumors were generated by subcutaneous injection of PC-3 cells into SCID mice. We treated the mice with vehicle control or the PARP inhibitor and found that olaparib significantly inhibited the growth of DTX3L-expressing tumors compared with EV control tumors (Fig. 7g, h). These data indicate that DTX3L overexpression sensitizes prostate cancer cells to PARP inhibitors in vitro and in vivo.

## Discussion

Targeting DNA damage response pathways has emerged as a promising anticancer strategy over the past decade; one case in which we have good insight is the synthetic lethality of PARP inhibitors and genetic HR deficiency caused by BRCA1 or BRCA2 mutations[34–36]. However, results from clinical studies have shown that genetic HR deficiency does not appear to be the only indicator of PARP inhibitor sensitivity in patients[36–40], emphasizing that additional biomarkers are urgently needed to guide PARP inhibitor usage in the clinic. Apparently, there are other mechanisms leading to HR deficiency. In our previous study, we demonstrated that TRABID overexpression contributes to HR deficiency and sensitizes prostate cancer cells to synthetic lethality by PARP inhibitors[11], indicating that protein posttranslational modifications are also important sources of HR deficiency. It was recently reported that TIRR is critical for favoring DNA repair by HR over NHEJ[12–16]. However, few TIRR mutations were found in patient samples from TCGA, suggesting that posttranslational modification of TIRR may result in sensitivity to PARP inhibitors in cancers. In the present study, we demonstrate that TIRR is degraded in the cytoplasm after being ubiquitinated by DTX3L upon DNA damage. Although the *TIRR* mRNA expression level was increased in prostate cancer, the protein was almost absent due to high expression of DTX3L (Supplementary Fig. 5a). We further show that DTX3L overexpression contributes to HR deficiency and sensitizes prostate cancer cells to synthetic lethality by PARP inhibitors (Figs. 6, 7). On the basis of our findings, we speculate that DTX3L and XPO1 facilitate TIRR nuclear export and degradation upon DNA damage (Fig. 8a) and that elevated

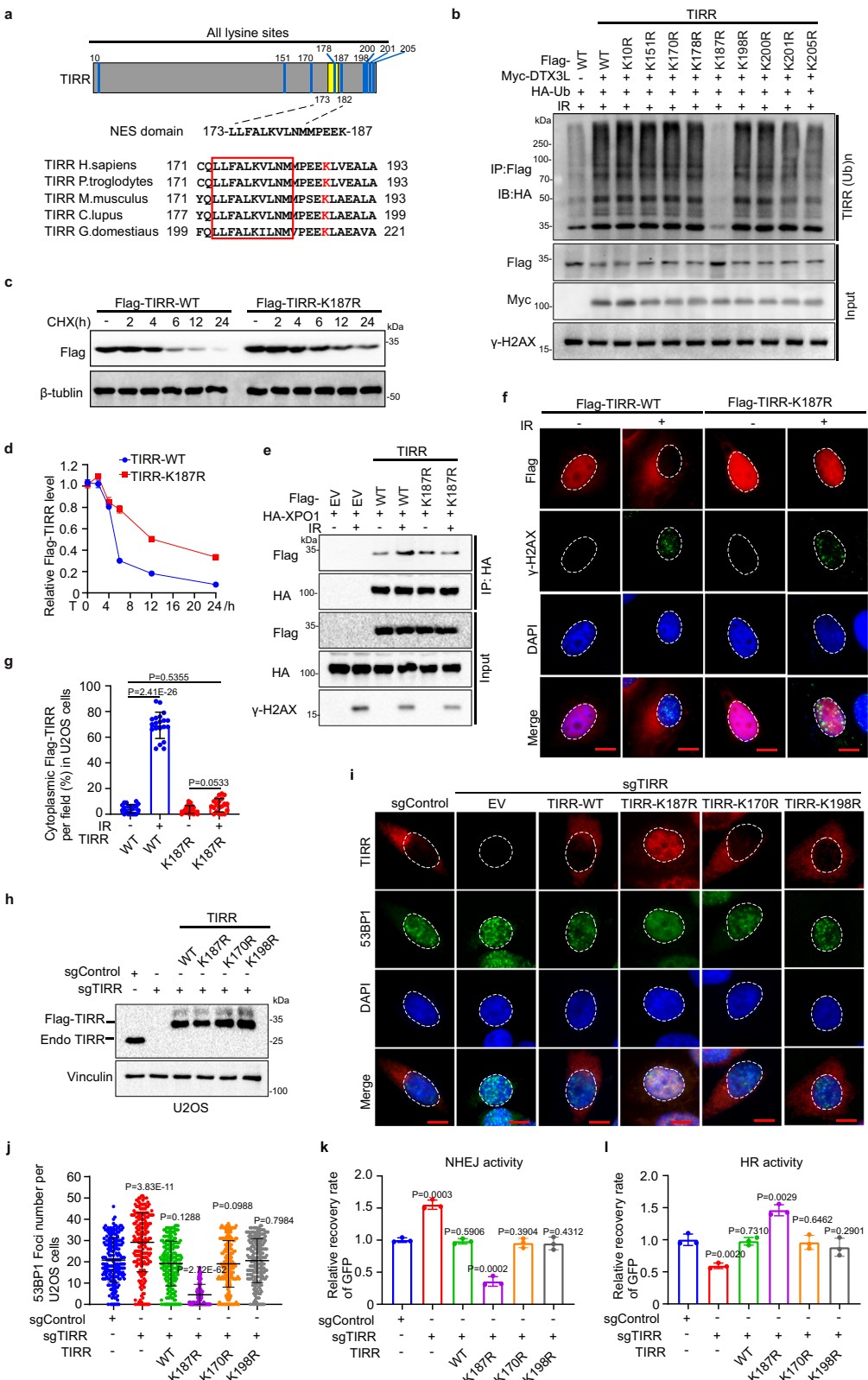

DTX3L expression induces HR deficiency by promoting TIRR degradation and subsequently sensitizes cancer cells such as prostate cancer cells to synthetic lethality by PARP inhibitors (Fig. 8b).

Nucleocytoplasmic transport of macromolecules is essential to cellular homeostasis and is regulated by the nuclear pore complex and transport proteins that mediate trafficking of molecules across the nuclear envelope[27,41,42]. Exportin-1 (XPO1; previously known as CRM1) is

responsible for the nucleocytoplasmic transport of hundreds of proteins and multiple RNA species. XPO1 recognizes the NESs of cargo proteins[22–27]. However, not all cargo proteins harbor nuclear export signals, and ubiquitination has been reported to facilitate protein export by XPO1 in the absence of an NES[43]. In the present study, we demonstrate that TIRR is exported by XPO1 with the help of DTX3L. Although TIRR harbors an NES to which XPO1 binds, the TIRR-XPO1

**Fig. 5 | Ubiquitination of TIRR at lysine-187 by DTX3L dictates NHEJ pathway activity and PARP inhibitor sensitivity. a** Homologous amino acid sequence alignments of NES motif and all the lysine residues (in blue). **b** In vivo poly-ubiquitination assay of whole-cell lysates from 293T cells transfected with HA-Ub, Myc-DTX3L, Flag-TIRR-WT, or mutants after MG132 treatment post-IR (8 Gy). WB analysis (**c**) and quantification (**d**) of Flag-TIRR protein in whole-cell lysates from 293T cells transfected with Flag-TIRR-WT or K187R mutant after CHX treatment. Data were shown as the mean ± SD from three biological replicates ($n = 3$). **e** Co-IP of whole-cell lysates from 293T cells transfected with HA-XPO1, Flag-TIRR-WT, or K187R mutant, harvested after IR (8 Gy). IFC (**f**) and quantification (**g**) of cytoplasmic Flag-TIRR-WT or K187R mutant in U2OS cells transfected with indicated constructs after IR (8 Gy). Scale bar, 10 μm. Data were shown as the mean ± SD from 20 fields (>200 cells, $n = 20$) from three biological replicates. Two-tailed unpaired Student's $t$-test; $P$ values based on the order of appearance: 2.41E-26, 0.5355,

0.0533. **h–j** Control or TIRR knockout U2OS cells were transfected with EV, Flag-TIRR-WT or mutants and treated with IR (8 Gy), followed by WB analysis in whole-cell lysates (**h**) and IFC of 53BP1(**i**). 53BP1 foci was quantified (**j**). Scale bar, 10 μm. Data were shown as the mean ± SD from three biological replicates ($n = 200$). Two-tailed unpaired Student's $t$-test; $P$ values based on the order of appearance: 3.83E-11, 0.1288, 2.75E-62, 0.0988, 0.7984. Analysis of NHEJ (**k**) and HR (**l**) activities in control or TIRR knockout U2OS cells transfected with HR or NHEJ reporter constructs in combination with EV, Flag-TIRR-WT, or mutants. Data were shown as the mean ± SD of three independent experiments ($n = 3$). Two-tailed unpaired Student's $t$-test; $P$ values based on the order of appearance: 0.0003, 0.5906, 0.0002, 0.3904, 0.4312 (**k**); 0.0020, 0.7301, 0.0029, 0.6462, 0.2901 (**l**). Source data are provided as a Source Data file. Similar results for (**b**, **c**, **e**, **h**) panels were obtained in three independent experiments.

interaction is weak in the absence of DTX3L-mediated TIRR ubiquitination. We demonstrate that DTX3L ubiquitinates TIRR at lysine-187, which is near the NES, to augment XPO1-mediated TIRR nuclear export upon DNA damage. One possible explanation is that the ubiquitin chain of TIRR may form a clip with the NES to stabilize the TIRR-XPO1 interaction. However, structural studies are needed to prove this point. We also noticed that the lysine-187 residue of TIRR, its ubiquitination site, was mutated to arginine in patient samples from TCGA, indicating that TIRR could be mutated in cancer and that TIRR mutation could contribute to disease progression.

In summary, we demonstrate that TIRR is precisely regulated in response to DNA damage. TIRR is translocated to the cytoplasm by XPO1-mediated nuclear export and degraded by DTX3L-mediated ubiquitination upon DNA damage. Ubiquitination of TIRR at lysine 187 by DTX3L is necessary for NHEJ pathway activity and influences PARP inhibitor sensitivity by facilitating XPO1-mediated TIRR nuclear export and degradation upon DNA damage. We further show that TIRR expression is decreased by DTX3L overexpression in prostate cancer, which inhibits HR and promotes chromosomal instability. Accordingly, we demonstrate that DTX3L overexpression sensitizes prostate cancer cells to synthetic lethality by PARP inhibitors, highlighting that DTX3L overexpression indicates sensitivity to PARP inhibitors in cancers such as prostate cancer.

## Methods

Further information and requests for resources and reagents should be directed to and will be fulfilled by the corresponding authors.

### Cell culture and transfections

293T and U2OS cells were cultured in Dulbecco's modified Eagle's medium (DMEM) supplemented with 10% FBS. PC-3 cells were cultured in RPMI 1640 medium supplemented with 10% FBS. The cells were maintained in a 37 °C humidified incubator containing 5% $CO_2$. The cells were routinely checked for mycoplasma infection and found to be negative. Transfection was performed with Lipofectamine 2000 (for plasmid transfection), polyethyleneimine (PEI) (for plasmid transfection) or Lipofectamine RNAiMAX (for siRNA transfection) according to the manufacturer's instructions. pTsin-DTX3L, lenticrisprV2-XPO1 or lenticrisprV2-DTX3L, and viral packaging plasmids were transfected into 293T cells. The virus-containing medium was collected 48 h or 72 h after transfection. PC-3 and U2OS cells were infected with viral particles using polybrene (8 μg/ml) and were then selected in growth medium containing puromycin (1.5 μg/ml).

### Generation and treatment of prostate cancer xenografts in mice

Six-week-old male SCID mice were bred in-house and used for animal experiments. All mice were housed under maintained under pathogen-free conditions at room temperature on a 12 h light/dark cycle and given access to food and water ad libitum. For studies of tumors treated with the PARP inhibitor olaparib, PC-3 cells ($5 \times 10^6$) infected

with lentivirus expressing pTsin-vector or pTsin-DTX3L were injected s.c. into the right flank of mice. The mice were injected intraperitoneally with vehicle or olaparib (50 mg/kg) daily. The volume of the xenografts was measured every 5 days for 30 days and calculated using the formula $0.5 \times$ length (L) $\times$ width (W)$^2$. The allowed maximal tumor size is 2 cm in any direction based on the institutional tumor production policies and none of the tumors exceeded this size at any point. Upon completion of the measurement, the tumors were harvested for imaging. The experiments with mice were conducted according to protocols approved by the Rules for Animal Experiments published by the Chinese Government and approved by the Ethics Committee of the First Affiliated Hospital of Xi'an Jiaotong University (Xi'an, China).

### Antibody and chemicals

Primary antibodies used include NUDT16L1 (Sigma-Aldrich, #HPA044186, 1:1000), 53BP1 (Abcam, #ab36823, 1:1000), 53BP1 (Santa Cruz, #sc-515841, 1:500), XPO1 (Cell Signaling, #46249,1:1000), Vinculin (Santa Cruz, #sc-73614, 1:1000), Lamin B1 (ABclonal, #A16909, 1:1000), β-tubulin (ABclonal, #AC021, 1:1000), DTX3L (Santa Cruz, #sc-514776, 1:1000), Myc (Santa Cruz, #sc-40, 1:1000), Flag (Cell Signaling, #8146, 1:1000), Flag-Alexa Fluor 594 (MBL, #M185-A59, 1:1000), HA (Cell Signaling, #3724, 1:1000), Ubiquitin (Cell Signaling, #14049, 1:1000), Phospho histone H2A.X (S139) (Cell Signaling, #9718, 1:1000), and Phospho histone H2A.X (S139) (Cell Signaling, #80312 S, 1:1000). Second antibodies were Rabbit IgG (H+L), FITC (Xi'an Zhuangzhi Biotechnology Co., Ltd, #EK023, 1:500), Rabbit IgG (H + L), Cy3 (Xi'an Zhuangzhi Biotechnology Co., Ltd, #EK022, 1:500), Mouse IgG (H + L), FITC (Xi'an Zhuangzhi Biotechnology Co., Ltd, #EK013, 1:500), Mouse IgG (H + L), Cy3 (Xi'an Zhuangzhi Biotechnology Co., Ltd, #EK012, 1:500), Rabbit IgG (Abclonal, #AS014, 1:5000), and Mouse IgG (Abclonal, #AS003, 1:5000). The following chemicals were used: Cisplatin (Selleck, #S1166), CPT (APExBIO, #A2877), KPT-330 (Selleck, #S7252), Olaparib (Selleck, #S1060), MLN4924 (Selleck, #S7109) and MG132 (Sigma-Aldrich, #133407-82-6).

### RNA interference and sgRNA-mediated gene deletion

A nonspecific control small interfering RNA (siRNA) and siRNAs for CtIP and 53BP1 were purchased from GE Dharmacon. The sequences of the siRNAs were as follows:

siCtIP#1 5′-GCUAAAACAGGAACGAAUC-3′;
siCtIP#2 5′-UCCACAACAUAAUCCUAAU-3′.
si53BP1#1 5′-GAGCUGGGAAGUAUAAAUU-3′;
si53BP1#2 5′-GGACUCCAGUGUUGUCAUU-3′.
The sequences of sgRNAs are as follows:
sgXPO1#1 5′-CACCGGCTAACATTGTCATAATTGC-3′;
sgXPO1#2 5′-CACCGTCAACTGTGTCAGTTTGTAA-3′;
sgDTX3L#1 5′- CACCGGATTTCACCTCAAGTCGATC-3′;
sgDTX3L#2 5′- CACCGGGAGGCGAATTAACTCTCCT-3′;
sgTIRR#1 5′-CACCGGCGGTGCACTCGCGCGACCA-3′;
sgTIRR#2 5′-CACCGGGTGCGCCACGACGCGGTGT -3′.

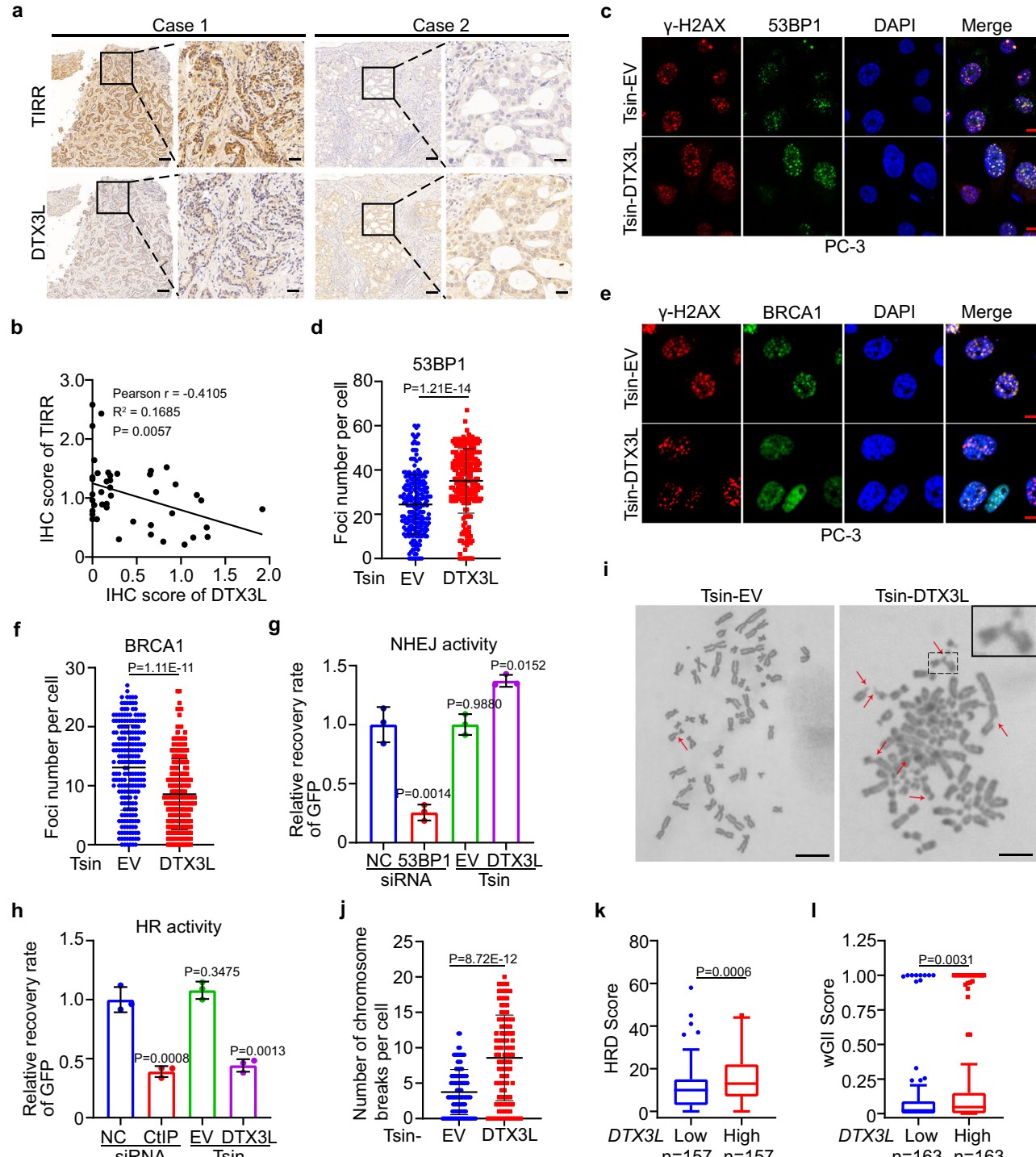

**Fig. 6 | DTX3L overexpression impairs HR activity and promotes chromosomal instability in prostate cancer. a** Representative IHC staining images of TIRR and DTX3L protein in prostate cancer samples (*n* = 44). Scale bar in 10 X fields: 400 μm; Scale bar in 40 X fields: 100 μm. **b** Pearson correlation analysis between DTX3L and TIRR IHC score in tumor tissues. Pearson *r* = −0.4105; Two-sided test; *P* = 0.0057. **c**–**f** PC-3 cells infected with lentivirus expressing EV or DTX3L were exposed to IR (8 Gy) followed by IFC of 53BP1 (**c**) and BRCA1 (**e**). The average foci number (**d**, **f**) per cell were quantified. Scale bar, 10 μm. Data were shown as the mean ± SD from three biological replicates (*n* = 200). Two-tailed unpaired Student's *t*-test; *P* = 1.21E-14 (**d**); *P* = 1.11E-11 (**e**). **g**, **h** PC-3 cells were transfected with HR or NHEJ reporter in combination with EV/DTX3L or small interfering RNA (siRNA) targeting CtIP or 53BP1. NHEJ (**g**) and HR (**h**) and activities were measured. Data were shown as the mean ± SD of three independent experiments (*n* = 3). Two-tailed unpaired Student's

*t*-test; *P* values based on the order of appearance: 0.0014, 0.9980, 0.0152 (**g**); 0.0008, 0.3475, 0.0013 (**h**). **i**, **j** PC-3 cells infected with lentivirus expressing EV, or DTX3L were treated with CPT (1 μM) for 24 h. Cells were harvested for karyotyping, and chromosome breaks were quantified. Representative images were shown in (**i**) and quantitative data are shown in (**j**). Scale bar, 10 μm. Data were shown as the mean ± SD from three biological replicates (*n* = 100). Two-tailed unpaired Student's *t*-test; *P* = 8.72E-12. Analysis of homologous recombination deficiency (HRD) score (**k**), weighted genome integrity index (wGII) score (**l**) in prostate cancer samples (see "Methods" section). The box outlines show the 25th and 75th percentiles, and the solid lines show the median value and the whiskers extending to the most extreme data point that is no more than 1.5 times the interquartile range. Two-sided Wilcoxon tests; *P* = 0.0006 (**k**); *P* = 0.0031 (**l**). Source data are provided as a Source Data file.

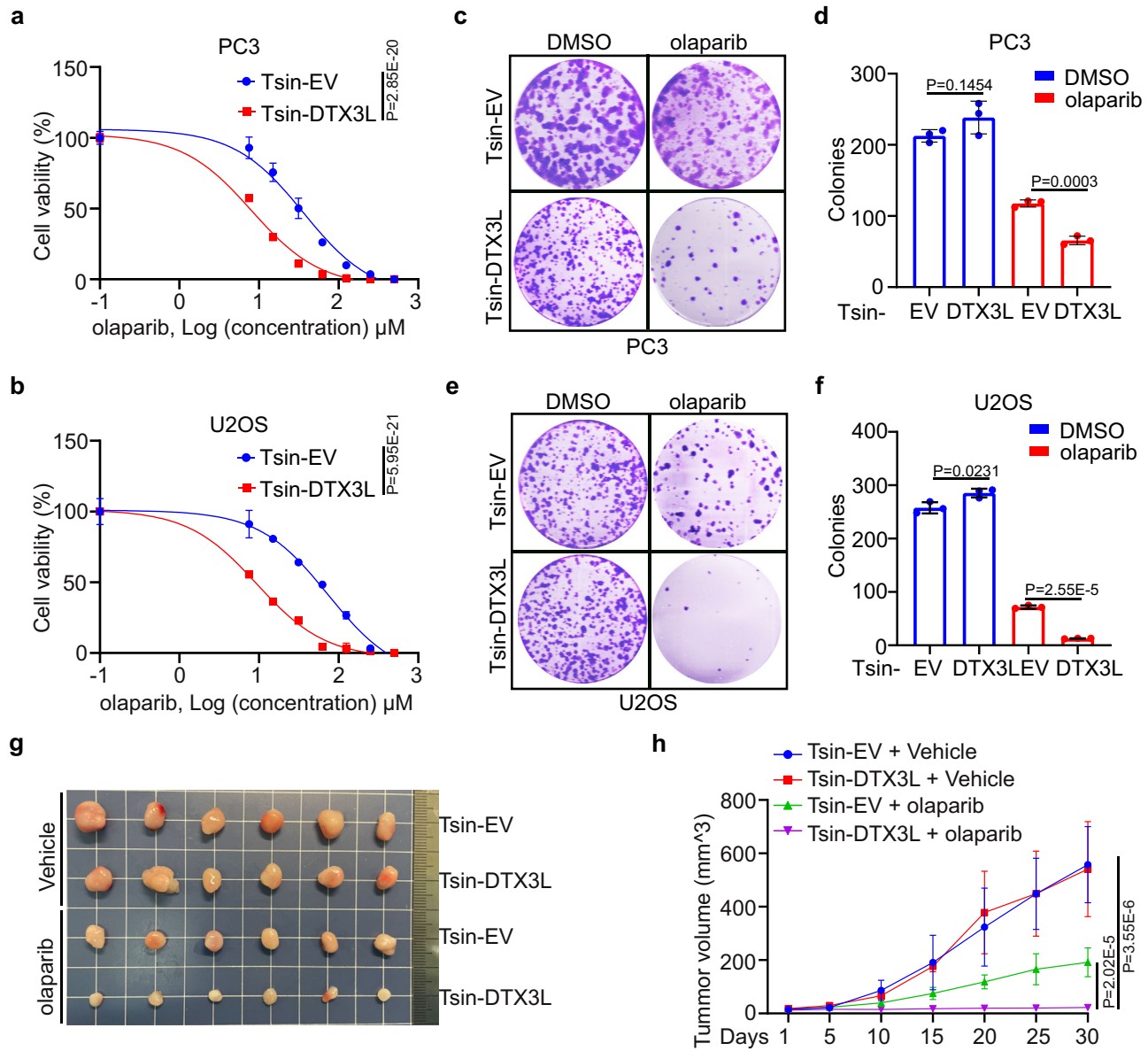

**Fig. 7 | DTX3L expression sensitizes prostate cancer cells to synthetic lethality by PARP inhibitors.** Dose-response survival curves for PC-3 (**a**) and U2OS (**b**) cells expressing either EV or DTX3L, exposed to increasing concentrations of olaparib. Data were shown as the mean ± SD of three independent experiments (*n* = 3). Two-way ANOVA; *P* = 2.85E-20, F = 385.1476 (**a**); *P* = 5.95E-21, F = 488.8113 (**b**). Colony formation assays in PC-3 (**c**, **d**) and U2OS (**e**, **f**) cell lines infected with lentivirus expressing EV or DTX3L. The number of colonies was counted. Representative colonies are shown in (**c**, **e**), with quantification data shown in (**d**, **f**). Data were presented as the mean ± SD of three independent experiments (*n* = 3). Two-tailed

unpaired Student's *t*-test. *P* values based on the order of appearance: 0.1454, 0.0003 (**d**); 0.0231, 2.55E-5 (**f**). **g**, **h** PC-3 cells infected with lentivirus expressing EV or DTX3L were injected into the right flank of SCID mice and treated with vehicle or olaparib (50 mg/kg). Tumor growth was measured every 5 days for 30 days. Tumors in each group at day 30 were harvested, photographed and shown in (**g**). Data in (**h**) are shown as mean ± SD (*n* = 6). Comparing the size of tumors in different groups at day 30. Two-tailed unpaired Student's *t*-test. *P* values based on the order of appearance: 2.02E-5, 3.55E-6. Source data are provided as a Source Data file.

## Plasmids and mutagenesis

The Flag-NUDT16L1 construct was purchased from MiaolingBio, China (no. P28448). HA-XPO1 was obtained from B. Ma. The Myc-DTX3L construct was purchased from MiaolingBio, China (no. P30656). The HA-DTX3L construct and the mutants of Flag-NUDT16L1, HA-XPO1, and HA-DTX3L were generated using the KOD Plus Mutagenesis Kit (TOYOBO, #SMK-101) following the manufacturer's instructions. The LenticrisprV2 plasmid was purchased from Addgene (no. 52961).

## Immunoblotting and immunoprecipitation

Cells were lysed in RIPA buffer (50 mM Tris-HCl (pH 7.5), 150 mM NaCl, 1% SDS, 0.5% sodium deoxycholate, and 1% NP-40) supplemented with

protease inhibitors and phosphatase inhibitors for 10 min. The lysates were sonicated for 1 min on ice in an ultrasonic cell disruptor and cleared by centrifugation at 16,000 × *g* for 10 min at 4 °C. The supernatant was stored at −20 °C until blotting. The protein concentrations were quantified by a Pierce Bicinchoninic Acid (BCA) Protein Assay Kit. The protein samples were mixed with 5x SDS loading buffer (250 mM Tris-HCl, pH 6.8, 10% SDS, 25 mM β-mercaptoethanol, 30% glycerol, and 0.05% bromophenol blue) and boiled for 5 min. Equal amounts of protein samples were then subjected to SDS–PAGE and transferred to nitrocellulose membranes. The membranes were blocked with 5% milk for 1 h at room temperature and incubated with primary antibody at 4 °C overnight. Then, the membranes were incubated with secondary

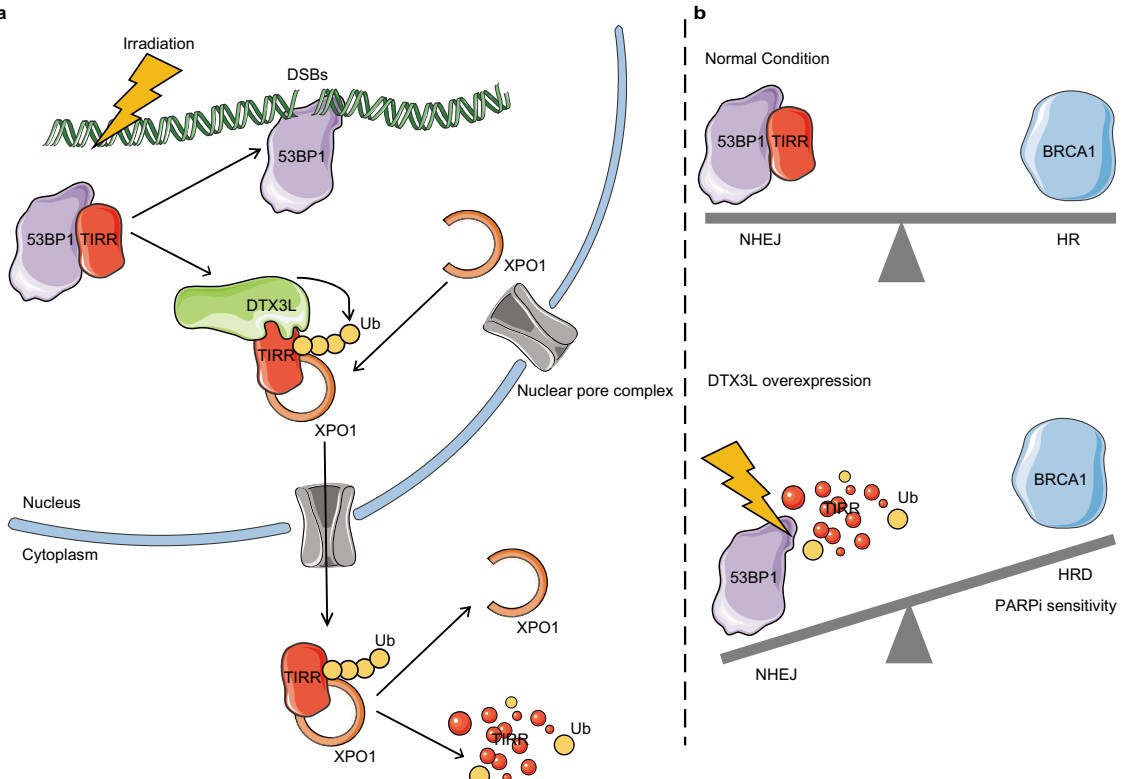

**Fig. 8 | Working model illustrating how DTX3L regulates TIRR stability by promoting TIRR ubiquitination, favoring NHEJ over HR and sensitizing cells to synthetic lethality by PARP inhibitors.** Under normal conditions, TIRR negatively regulates 53BP1 by forming a complex with 53BP1. Upon DNA damage, TIRR is translocated to the cytoplasm by XPO1-mediated nuclear export and degraded by DTX3L-mediated ubiquitination (**a**). However, prostate cancer-associated DTX3L overexpression promotes the nuclear export and degradation of TIRR, which consequently leads to increased accumulation of 53BP1 at DSBs; the interaction of 53BP1 with chromatin at DSBs causes NHEJ to be favored over HR and sensitizes cells to the lethal effects of PARP inhibitors (**b**). Figure 8 was partly generated using images from Servier Medical Art by Servier, licensed under a Creative Commons Attribution 4.0 unported license (https://creativecommons.org/licenses/by/4.0/).

antibody for 1 h at room temperature. The protein bands were visualized by ECL reagents, imaged on a ChemiDoc XRS+ Imager, and analyzed with Image Lab Software.

For immunoprecipitation, the cells were lysed in IP buffer (50 mM Tris-HCl, pH 7.4, 150 mM NaCl, 1% NP-40, and 1% sodium deoxycholate) supplemented with protease inhibitors and phosphatase inhibitors. The cell lysates were centrifuged for 10 min at 16,000 × $g$ at 4 °C, and the supernatants were incubated with primary antibody-conjugated protein A/G beads or HA-/Flag-conjugated agarose with rotation at 4 °C overnight. The next day, the beads were washed extensively with IP buffer, and the proteins were extracted by boiling at 95 °C for 5 min. Finally, the proteins were separated by SDS–PAGE for western blotting.

### Nuclear-cytoplasmic fractionation

Trypsin-dispersed cells were neutralized and collected by centrifugation using a microcentrifuge at 200 × $g$. The cells were resuspended gently with PBS. One-third of the volume of resuspended cells was used for immunoblot analysis of whole-cell lysates as described above. The other 2/3 of the volume of resuspended cells was centrifuged, and the pellet was resuspended in CE buffer (10 mM HEPES, 60 mM KCl, 1 mM EDTA, 0.075% (v/v) NP-40, 1 mM DTT, and 1 mM PMSF). The cytoplasmic fraction was lysed after incubation on ice for 3 min with gentle flicking. The nuclei was pelleted by microcentrifugation at 200 × $g$ for 4 min. The supernatant, which contained the cytoplasmic fraction, was transferred to a clean tube. After gently washing with cold PBS and centrifugation at 200 × $g$ for 4 min, the nuclear pellet was resuspended in NE buffer (20 mM Tris-HCl, 420 mM NaCl, 1.5 mM MgCl2, 0.2 mM EDTA, 1 mM PMSF, and 25% (v/v) glycerol, adjusted to pH 8.0). The nuclear lysates were vortexed and sonicated for 1 min in

an ultrasonic cell disruptor. After incubating the samples for 10 min, the cytoplasmic and nuclear fractions were spun at maximum speed for 10 min to pellet any nuclei. The supernatants were transferred separately to clean tubes and then subjected to immunoblotting.

### Fluorescence immunohistochemistry

PC-3 or U2OS cells were seeded on 13-mm glass coverslips. The coverslips were fixed with 4% paraformaldehyde for 15 min and permeabilized with 0.2% Triton for 10 min at room temperature. The samples were blocked with 5% BSA for 1 h, incubated with primary antibodies for 2 h, washed at least three times with 0.01% Tween 20 in PBS (PBS-T), and subsequently incubated with fluorophore-coupled secondary antibodies for 1 h. After three washes, the coverslips were mounted on glass slides using one drop of mounting medium containing 4′,6-diamidino-2-phenylindole (DAPI). Images were acquired with a confocal microscope.

For the quantification of 53BP1 or γ-H2AX foci, 10 randomly selected photographs of more than 200 cells were used for normalization of focus number and measurement of the integrated optical density with ImageJ (version 1.53, NIH). Approximately 200 cells were analyzed for each series of experiments.

### Ubiquitination assay

293T cells were transfected with HA-Ub, Flag-NUDT16L1, and other plasmids as indicated. Twenty-four hours after transfection, the cells were treated with 10 μM MG132 for 6 h and lysed with IP buffer on ice for more than 10 min. The lysate was sonicated and centrifuged for 15 min at 16,000 × $g$ at 4 °C, and the supernatant was incubated with Flag-conjugated agarose beads (Sigma–Aldrich) with rotation at 4 °C

overnight. The next day, the beads were washed extensively with IP buffer, and the proteins were extracted by boiling at 95 °C for 5 min. Finally, the proteins were separated by SDS–PAGE for western blotting.

## Tandem ubiquitin-binding entity (TUBE) pulldown assay

TUBE pulldown assay was performed according to a previously published procedure[44]. Briefly, cells were treated with 10 µM MG132 for 6 h, then lysed with TUBE lysis buffer (50 mM Tris-HCl, 0.15 M NaCl, 1 mM EDTA, 1% NP-40, 10% glycerol, pH 7.5) supplemented with additional inhibitors: 1,10-phenanthroline (o-PA) (5 mM), N-Ethylmaleimide (NEM) (5 mM), and PR-619 (100 µM). After centrifugation, the cell lysate was incubated with agarose-TUBE beads (LifeSensors, #UM-0402-1000) at 4 °C for 2 h. The agarose-TUBE beads were then washed three times with TBST and boiled in 1.5× SDS loading buffer. All samples were subsequently analyzed by SDS-PAGE for western blotting.

## Tissue microarray immunohistochemistry (IHC)

Prostate cancer tissue samples were obtained from the First Affiliated Hospital of Xi'an Jiaotong University (Xi'an, China). The protocol was approved by the Ethical Committee of the First Affiliated Hospital of Xi'an Jiaotong University. Informed consent was obtained from all patients. Formalin-fixed, paraffin-embedded tumor tissue samples were sectioned at 4 µm. The slides were stained with the indicated antibodies according to a standard IHC protocol. Images were taken using an Olympus microscopic camera and the corresponding software. Semiquantification of protein expression was performed by the use of scoring criteria. The proportion of stained cells (%) and staining intensity (0, no staining; 1, weak staining; 2 moderate staining; 3 strong staining) were assessed, and these values were then multiplied to yield a score ranging from 0 to 3. To maintain consistency, the same qualified pathologist interpreted all IHC data.

## Quantitative real-time polymerase chain reaction (real-time qPCR)

Total RNA (0.5–1 mg) was isolated using RNAfast 200 reagents according to the manufacturer's protocol, and the RNA concentration was quantified by a multiwavelength microplate reader at 260 nm. The RNA was reverse transcribed with PrimeScript Real-Time Master Mix. To determine the relative mRNA level, real-time qPCR was performed using 2x SYBR Green qPCR Master Mix, and gene expression was normalized to the expression of 18S. The DDCt method was used to evaluate the relative expression level of the indicated genes. The sequences of the primers are listed in Supplementary Table 1.

## HR and NHEJ reporter assay

Cells were transfected with siControl, siCtiP, si53BP1, Tsin-EV, or Tsin-DTX3L separately or in combination with HR (pDR-GFP)- or NHEJ (pPEM1-Ad2-EGFP)-reporter constructs and an expression vector for the restriction enzyme I-Sce I[10]. GFP expression induced by the positive control plasmid was used to normalize the electroporation efficiency. The cells were grown for 48 h and processed for flow cytometry.

## HRD score and wGII analysis

The *DTX3L* RNA expression profiles as FPKM of the Cancer Genome Atlas (TCGA) prostate cancer (PRAD) cohort were downloaded from the Genomic Data Commons (GDC) (https://portal.gdc.cancer.gov/). The tumor weighted GII (wGII) score was calculated as previously defined[33]. Specifically, the genomic integrity index (GII) was calculated as the fraction of the genome with aberrant copy numbers (more than 0.3 different) relative to the baseline ploidy. The wGII score was calculated as the mean fraction of GII across all chromosomes. The homologous recombination deficiency (HRD) score was collected from published research[32]. Specifically, the HRD score is a sum of three components of genome scarring scores, including loss of heterozygosity score, large-scale state transitions score, and number of telomeric allelic imbalances score[31].

## Karyotype analysis

PC-3 cells were treated with 1 µM CPT for 24 h and colcemid for 1 h before harvest. The cells were washed two times in PBS and then resuspended in 0.075 M KCl at 37 °C for 45 min. The cells were fixed with fixative (3:1 methanol: glacial acetic acid) twice for 15 min each time. Small drops of cell suspensions were placed onto a slide surface and stained with Diff-Quick stain for 1 min. Approximately 100 cells with well-spread chromosomes were photographed and analyzed in each group.

## 3-(4,5-Dimethylthiazol-2-yl)-2,5-diphenyltetrazolium bromide (MTT) assay and clonogenic assay

To assess cell viability, cells were plated at a density of 3000 cells/well in 96-well plates. After 24 h, the cells were treated with different concentrations of drugs for 48 h. Before harvest, we replaced the medium with fresh medium containing MTT (0.5 mg/ml), and then the cells were incubated for 2 h. Then, we aspirated the medium, added DMSO, and shook the plate on an orbital shaker for 10 min. The OD value was read at a wavelength of 570 nm. Three measurements for each sample were averaged, and the background signal (medium) was subtracted, and then quantitative analysis was performed.

For the clonogenic survival assay, an appropriate number of cells was plated in six-well plates according to the drug dose. After 24 h, the cells were treated with DMSO or different doses of drugs. Twelve days later, the colonies were fixed and stained with 0.5% crystal violet (w/v) for 1 h. The number of colonies in each group was counted and analyzed.

## Isothermal titration calorimetry (ITC) assay

XPO1 protein was expressed in E. coli and purified via affinity chromatography, while the NES peptide (173–182 aa of TIRR) was synthesized using solid-phase peptide synthesis and purified by preparative high-performance liquid chromatography. Prior to ITC experiments, the sample cells and injection syringe were cleaned as per the manufacturer's protocol. The sample cell was rinsed multiple times with buffer and loaded with 0.2 ml of protein solution (concentration ~20 µM), ensuring no bubbles formed. The syringe was filled with 0.04 ml of peptide solution (concentration ~200 µM). ITC experiments were conducted at 25 °C, using a small volume (0.4 µl) for the first injection, followed by 19 injections of 2 µl each. After the experiment, the ITC data was analyzed using MicroCal iTC200 system (Malvern Panalytical, Malvern, United Kingdom) and MicroCal PEAQ-ITC Analysis Software (version 1.41, Malvern Panalytical).

## Mass spectrometry analysis

Liquid chromatography–mass spectrometry (LC–MS) analysis was performed according to a previously published procedure[45]. 293T cells were transfected with empty vector (EV) or Flag-tagged TIRR. After 48 h, the cells were lysed with IP buffer and immunoprecipitated with Flag-conjugated agarose beads (Sigma–Aldrich, #A2220). The bound proteins were eluted with 2% SDS and digested overnight with sequencing-grade modified trypsin (Promega, #V5111). The obtained peptides were analyzed using a nanoflow EASY-nLC 1200 system (Thermo Fisher Scientific, Odense, Denmark) coupled to an Orbitrap Exploris480 mass spectrometer (Thermo Fisher Scientific, Bremen, Germany). The data were searched against the UniProt human protein database (75,004 entries, download on 07-01-2020) using Protein Discoverer (version 2.4.1.15, Thermo Fisher Scientific) and Mascot (version 2.7.0, Matrix Science).

## Quantification and statistical analysis

All the experiments were repeated at least three times unless otherwise indicated. Graphs were generated using GraphPad Prism 8 (GraphPad, Inc.) or Microsoft Office Excel 2010. All numerical data are presented as the mean ± SEM or mean ± SD as appropriate. Differences between groups were analyzed by t-tests, two-way ANOVA, or two-sided Wilcoxon tests with GraphPad Prism 8.

## Reporting summary

Further information on research design is available in the Nature Portfolio Reporting Summary linked to this article.

## Data availability

The raw mass spectrometry proteomics data have been deposited to the ProteomeXchange Consortium (http://proteomecentral.proteomexchange.org) via the iProX partner repository[46] with the dataset identifier PXD052392 and PXD055432. All remaining data needed to evaluate the conclusions in the paper are provided in the paper and its Supplementary Information files. Source data are provided with this paper.

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

## Acknowledgements

This work was supported, in part, by grants from the National Natural Science Foundation of China (81925028 to L.L., 82230097 to L.L., and 82173037 to J.M.) and National Key R&D Program of China (2023YFC3404104 to L.L.). We thank the Instrumental Analysis Center of Xi'an Jiaotong University for their help in characterizations.

## Author contributions

Q.Y. and J.M. have contributed equally to this work. J.M. and L.L. (corresponding author) conceived the study. Q.Y., Z.W., L.L., T.L., B.W., L.Z., and Y.L. performed experiments, data collection, and analysis. S.X., K.W., Y.J., B.M., Y.F., J.L., and Y.G. provided conceptual advice. L.L. (corresponding author), H.H., and J.M. supervised the study and wrote the manuscript.

## Competing interests

The authors declare no competing interests.
