## [Peer Review file · Nature Communications]

DTX3L-mediated TIRR nuclear export and degradation regulates DNA repair pathway choice and PARP inhibitor sensitivity

Corresponding Author: Professor Lei Li

Version 1:

Reviewer comments:

Reviewer #1

(Remarks to the Author)

The manuscript entitled “Dual action of DTX3L on TIRR nuclear export and degradation regulates DNA repair pathway choice and PARP inhibitor sensitivity” by Ma et al., presents evidence indicating that XPO1 facilitates TIRR nuclear export and degradation in response to DNA damage. This process is mediated by the E3 ligase DTX3L. The authors further proposed DTX3L as a potential biomarker for predicting sensitivity to PARP inhibitors in cancers. Some of the data presented are robust and effectively support the author's hypothesis. However, there are several important points that are insufficiently supported by the presented data, and these critical issues must be thoroughly addressed to justify a publication in Nature Communications.

Major issues

1. The primary focus of this manuscript is the translocation of TIRR from the nucleus to the cytoplasm. Hence, it is imperative for the authors to specify in the figure legends which portion of the cell extract was used for the IP and western blotting.
2. In Figure 1f and S1e, it is hard to discern whether the 53BP1 foci are increased after IR due to low quality of the images. This also happened in other images, probably two cells are sufficient to better representative images. In Figure 1f and S1e, discerning whether the 53BP1 foci increase after IR is challenging due to the low quality of the images. This issue was observed in other images as well. Perhaps limiting the images of two cells could provide clearer and more representative results.
3. It would be beneficial to identify the key residues on XPO1 necessary for the XPO1/TIRR interaction. This information would help determine whether the XPO1/TIRR interaction is required for TIRR nuclear export.
4. The authors demonstrated the interaction of TIRR-XPO1. It remains unclear whether this XPO1/TIRR interaction is dose-dependent or time-dependent in response to DNA damage.
5. In Fig S2b, the H2AX expression is reduced in the nucleus compared to the whole cell extract, whereas this alteration is not observed in Figure S3j.
6. The author claimed that “We noticed from our IFC data that the TIRR signal in the cytoplasm was weaker than that in the nucleus after IR (Fig. 2f, g)”. However, there is no evidence supporting this claim, thus raising the question about the justification for conducting the experiment presented in Figure 3a.
7. MLN4924 is a NEDD8-activating enzyme inhibitor. The rationale for using this drug in the experiment remains unclear, and consequently, the conclusion drawn from it lacks justification.
8. TIRR ubiquitination can be detected at the endogenous level in response to DNA damage.
9. The author claimed that “Notably, there was more DTX3L binding to TIRR in nuclear than that in cytoplasm, indicating the DTX3L-mediated TIRR ubiquitination happens before nuclear export of TIRR (Fig. 4l)”. In fact, there is a higher abundance of TIRR protein in the cytoplasm compared to the nucleus after IR. Thus, the weak DTX3L/TIRR interaction in the cytoplasm might be attributed to TIRR ubiquitination and subsequent degradation in the cytoplasm. Thus, the conclusion drawn from this observation is not well justified.
10. The authors demonstrated the interaction of TIRR-DTX3L. It remains unclear whether this DTX3L/TIRR interaction is dose-dependent or time-dependent in response to DNA damage.
11. It would be beneficial to confirm whether the RING domain of DTX3L is responsible for TIRR ubiquitination.
12. What is the rationale for using His-Ub in Fig S3l.
13. It would be preferable to identify the key ubiquitinated sites of TIRR using mass spectrometry analysis. Moreover, conducting additional experiments to confirm that DTX3L cannot degrade the TIRR K187A protein may provide further

insights.

14. For the NHEJ and HR reporter assays, it would be beneficial to show the efficiency of TIRR knockdown and plasmid transfection. These controls are crucial for quantifying the results of NHEJ and HR reporter assay.

15. There is a concern regarding the inconsistency between the protein levels and the foci of 53BP1 and BRCA1 in Figure S5c~5f.

16. DTX3L expression sensitizes prostate cancer cells to synthetic lethality by PARP inhibitors, However, there is no direct evidence linking the function of DTX3L to TIRR.

Minor issues

1. The authors should indicate the molecular weight of proteins for western blots.

3. The authors should provide more detailed descriptions in the figure legends. This would enhance readability and, consequently, improve the impact and reproducibility of the manuscript.

Reviewer #2

(Remarks to the Author)

In this manuscript, Ma et al. report their investigation into the regulation of TIRR, a protein that inhibits the DNA damage response protein 53BP1 and plays a key role in facilitating DNA double-strand break repair by homologous recombination. They demonstrate that, following DNA damage, TIRR is modified at lysine 187 (K187) by the E3 ubiquitin ligase DTX3L and exported from the nucleus to the cytoplasm through physical interaction with exportin-1 (XPO1). The nuclear export signal (NES) of TIRR was identified (motif M3 in the manuscript), with K187 adjacent to this NES. Polyubiquitinated TIRR undergoes degradation in the cytoplasm. The authors also show that overexpression of DTX3L in prostate cancer reduces TIRR levels, correlating with inhibition of homologous recombination and chromosomal instability. Reduced TIRR levels sensitize prostate cancer cells to synthetic lethality induced by PARP inhibitors, suggesting that DTX3L overexpression could be a marker for increased sensitivity to PARP inhibitors in prostate cancer. The manuscript is clearly written, well-organized with informative figures, and technically sound. Notwithstanding discrepancies with a preprint in bioRxiv that require attention (see below), the study by Ma et al. will be of great interest to researchers in the DNA repair field.

Issue requiring attention

A preprint posted in bioRxiv also reports a detailed characterization of the interaction between TIRR and XPO1 and investigates the TIRR export mechanism (<https://www.biorxiv.org/content/10.1101/2024.02.19.580988v1>). In this preprint, two nuclear export signals, not analyzed by Ma et al., were shown to mediate the interaction of TIRR with XPO1. Furthermore, the preprint indicates that mutations in the NES motif identified by Ma et al. (M3) do not affect the interaction of TIRR with XPO1. Consequently, the authors of the preprint excluded this specific site as an interaction motif for XPO1. These discrepancies are intriguing and should be addressed by Ma et al.

Reviewer #3

(Remarks to the Author)

I co-review this manuscript with one of the reviewers who provided the listed reports. This is part of nature communications initiative to facilitate the training in peer review and to provide appropriate recognition for Early Career Researcher who co-review manuscripts.

Reviewer #4

(Remarks to the Author)

The balance between homologous recombination (HR) and non-homologous end-joining (NHEJ) for repairing DNA double-stranded breaks (DSBs) has been well-established, and serves as one of the main regulatory mechanisms is through 53BP1 recruitment to the DSB sites and suppressing BRCA1-mediated HR. In this manuscript, Ma et al. uncovered an additional layer of 53BP1 regulation, which is through the subcellular localization changes in the 53BP1-interacting protein TIRR. The nuclear export of TIRR is triggered by ubiquitination of its TES domain in response to DNA damage, and this modification is required for XPO1 interaction to mediate TIRR export. Moreover, the authors identified the E3 ubiquitin ligase responsive to the ubiquitination as well as a ubiquitination site that plays a major role in the pathway. The data provided generally support their proposed model after some points are addressed. In particular, the exact subcellular location where TIRR degradation happens needs clarification, as detailed further below. Finally, the results from patient samples and animal models suggest the potential of targeting this pathway for PARP inhibitor sensitization. Although the overall novelty and the significance of this study are moderate, the well-prepared manuscript, and the robust, complete, and clear presentation of experimental data mostly compensate for this modest weakness.

Below are the major points for this study:

1. It's unclear whether the TIRR protein degradation happens in the nucleus or cytoplasm. Although the author proposed the degradation happens after translocation to cytosol, the evidence is insufficient. For example, did they see less protein degradation after DNA damage when knocking out XPO1? Is DTX3L only expressed in the nucleus? Additionally, inhibiting ubiquitination by MLN4924 not only induces nuclear accumulation but also increases the total protein level of TIRR (Figure 3e-g). Does this suggest that some degradation can also happen in the nucleus? The proteasome is known to be both cytoplasmic and nuclear, although perhaps mostly the former.
2. Regarding TIRR ubiquitination, the data presented in Figure 3 are mostly done by exogenous proteins. Did they confirm if

endogenous TIRR can also be ubiquitinated by DTX3L with endogenous ubiquitin pulldowns (i.e., a TUBE or Ub pulldown to complement Figure 4i)? This would strengthen the manuscript. Also, did they confirm the ubiquitination level of TIRR after treating with MLN4924 and MG132?

3. It would improve the manuscript if they rescued their DTX3L knockout with WT and a catalytically inactive mutant to further prove that the E3 ligase activity of DTX3L is ubiquitinating TIRR.

4. Figure 4e, the author should confirm whether the endogenous TIRR has a similar response to DTX3L overexpression and MG132 treatment, as compared to the exogenous TIRR.

5. The K187R mutant seems to have a dominant effect on the 53BP1-mediated DSB repair pathway choices. It would be interesting if at least one of the other mutants used in Figure 5b would behave more like wildtype TIRR, to further confirm the specificity of K187 ubiquitination.

6. Figure 6i-j, why does DTX3L expression induce more chromosome breaks? The authors need to further explain why this phenotype is associated with the shift of two DSB repair pathways. If not, what could be the possible reason?

Minor points:

7. In Figure 1c, why does the overall intensity of 53BP1 increase after DNA damage? The point that the authors want to make is that 53BP1 "foci" are increased upon DNA damage. However, the representative images did not show clear foci formation and the data looks more like just the overall intensity elevation. The authors need to clearly describe how they define the foci positive cells, and how they distinguish this from an overall intensity change. Also, the corresponding figure legend needs to be fixed, it's a co-staining of Flag and "53BP1", not with p-H2AX.

8. Figure 6e: it seems like the overall intensity of BRCA1 is decreased by DTX3L. Again, the author needs to clarify the changes they observed are reflected by foci number, not the overall intensity.

Reviewer #5

(Remarks to the Author)

Version 2:

Reviewer comments:

Reviewer #1

(Remarks to the Author)

My comments have been addressed in the revised version. I support the publication of this manuscript.

Reviewer #2

(Remarks to the Author)

The authors have adequately addressed my comments. Using quantitative binding assays (ITC) with TIRR peptides, they clearly demonstrate that only NES4 interacts with XPO1, whilst the other putative binding sequences (NES1, NES2, and NES3 peptides) show no affinity for XPO1. Co-IP of whole-cell lysates with HA-XPO1 and Flag-TIRR also support these findings.

The K_d value for NES4 peptide should be included on the ITC graph or in the figure legend. Additionally, it would make sense to incorporate these critical ITC and co-IP data into one of the figures in the manuscript.

Overall, this is a well-executed study that is suitable for publication in Nature Communications.

Reviewer #3

(Remarks to the Author)

Reviewer #4

(Remarks to the Author)

The authors have done a thorough job with addressing all of the previous concerns. I believe the manuscript should now be accepted.

Reviewer #5

(Remarks to the Author)

Authors' Response to Reviewers' Comments on Manuscript NCOMMS-24-19963A-Z

We thank the Editor and the Reviewers for the improvement of our manuscript.

Reviewer #1 (Remarks to the Author):

The manuscript entitled “Dual action of DTX3L on TIRR nuclear export and degradation regulates DNA repair pathway choice and PARP inhibitor sensitivity” by Ma et al., presents evidence indicating that XPO1 facilitates TIRR nuclear export and degradation in response to DNA damage. This process is mediated by the E3 ligase DTX3L. The authors further proposed DTX3L as a potential biomarker for predicting sensitivity to PARP inhibitors in cancers. Some of the data presented are robust and effectively support the author's hypothesis. However, there are several important points that are insufficiently supported by the presented data, and these critical issues must be thoroughly addressed to justify a publication in Nature Communications.

Response: We very much appreciate the reviewer for the positivity and enthusiasm about our study.

Major issues

1. The primary focus of this manuscript is the translocation of TIRR from the nucleus to the cytoplasm. Hence, it is imperative for the authors to specify in the figure legends which portion of the cell extract was used for the IP and western blotting.

Response: We thank the Reviewer for pointing this out. As suggested, we have carefully modified the description of cell extract in the revised legends, as shown in the manuscript: **Line 736 page 27, Lines 748, 755 page 28, Lines 774, 779, 781, 783, 785 page 30, Lines 796, 800, 801, 804, 805, 806, 808 page 32, Line 811 page 33 and Lines 822, 825 page 34, and Line 835 page 35.** So as shown in the supplementary materials: **Lines 33, 35 page 2, Lines 46, 47, 49, 51, 53, 55 page 3, Line 62 page 4, Lines 64, 67, 69, 71, 74, 80, 83, 86, 89, 91 page 5, Lines 101, 104 page 6, Lines 107, 109 page 7, Line 131 page 8 and Line 140 page 9.**

2. In Figure 1f and S1e, it is hard to discern whether the 53BP1 foci are increased after IR due to low quality of the images. This also happened in other images, probably two cells are sufficient to better representative images. In Figure 1f and S1e, discerning whether the 53BP1 foci increase after IR is challenging due to the low quality of the images. This issue was observed in other images as well. Perhaps limiting the images of two cells could provide clearer and more representative results.

Response: We thank the Reviewer for raising this great point. As showed in **Fig. 1a, f, Fig. 2f, g, Fig. 3k, m, Fig. 4m and Supplementary Fig. 1 a, e**, we have replaced the larger images with those depicting one or two cells throughout the manuscript for clearer representation.

3. It would be beneficial to identify the key residues on XPO1 necessary for the XPO1/TIRR interaction. This information would help determine whether the XPO1/TIRR interaction is required for TIRR nuclear export.

Response: We thank the Reviewer for these helpful suggestions. Structural studies have demonstrated that I521, L525, F561 and F572 in XPO1 as key residues forming a hydrophobic groove essential for NES recognition (*Nature*. 2009 PMID: 19339969; *Mol Cell*. 2004 PMID: 15574331). Consequently, we conducted a Co-IP assay using wild-type and mutant (I521A/L525A/F561A/F572A) XPO1. As demonstrated in the revised **Supplementary Fig. 2b and Lines 113-116 page 5**, these mutations abolished XPO1's binding to TIRR, indicating classical NES recognition is essential for XPO1/TIRR interaction.

4. The authors demonstrated the interaction of TIRR-XPO1. It remains unclear whether this XPO1/TIRR interaction is dose-dependent or time-dependent in response to DNA damage.

Response: We thank the Reviewer for raising these excellent points. As demonstrated in the revised **Supplementary Fig. 2c, d and Lines 117-120 page 5**, the interaction between TIRR and XPO1 was strengthened with higher IR doses and progressively increased over time following IR exposure, indicating the XPO1/TIRR interaction is both dose-dependent and time-dependent in response to DNA damage.

5. In Fig S2b, the γ H2AX expression is reduced in the nucleus compared to the whole cell extract, whereas this alteration is not observed in Figure S3j.

Response: We thank the Reviewer for pointing this out. We initially suspected that the observed alteration was due to the sensitivity of γ -H2AX protein levels to variations in independent cell fraction extraction experiments. To address this, we carefully repeated the experiments shown in revised **Supplementary Fig. 2e and Supplementary Fig. 3q**, taking extra care to maintain consistent experimental procedures. The new results were now consistent, allowing us to replace the previous data.

6. The author claimed that “We noticed from our IFC data that the TIRR signal in the cytoplasm was weaker than that in the nucleus after IR (Fig. 2f, g)”. However, there is no evidence supporting this claim, thus raising the question about the justification for conducting the

experiment presented in Figure 3a.

Response: This is an excellent point that we completely agree. To address this, we assessed the mean fluorescence intensity (MFI) of TIRR after IR treatment and observed a reduction in MFI post-IR (**Fig. 3a-d**). Consequently, we revised our original statement in the revised manuscript **Line 131 page 6** to “We noticed from our IFC data that the TIRR signal progressively decreased over time following IR exposure.

7. MLN4924 is a NEDD8-activating enzyme inhibitor. The rationale for using this drug in the experiment remains unclear, and consequently, the conclusion drawn from it lacks justification.

Response: We thank the reviewer for raising this point. To determine whether ubiquitination occurs before TIRR nuclear export, we used the E1 inhibitor MLN4924 to block TIRR ubiquitination and the proteasome inhibitor MG132 to prevent TIRR degradation. We have added this description in the revised manuscript **Lines 138-140 page 6**.

8. TIRR ubiquitination can be detected at the endogenous level in response to DNA damage.

Response: We appreciate the Reviewer’s valuable suggestion. In line with Reviewer #4’s recommendation, we incorporated an endogenous tandem ubiquitin-binding entity (TUBE) pulldown assay, which confirmed endogenous TIRR ubiquitination in response to DNA damage, as demonstrated in the revised **Fig. 4i**.

9. The author claimed that “Notably, there was more DTX3L binding to TIRR in nuclear than that in cytoplasm, indicating the DTX3L-mediated TIRR ubiquitination happens before nuclear export of TIRR (Fig. 4l)”. In fact, there is a higher abundance of TIRR protein in the cytoplasm compared to the nucleus after IR. Thus, the weak DTX3L/TIRR interaction in the cytoplasm might be attributed to TIRR ubiquitination and subsequent degradation in the cytoplasm. Thus, the conclusion drawn from this observation is not well justified.

Response: We agree with the Reviewer. As we have already demonstrated that TIRR ubiquitination occurs before nuclear export in **Fig. 3k-n**, further discussion on this point is unnecessary. Consequently, we have removed it in the revised version.

10. The authors demonstrated the interaction of TIRR-DTX3L. It remains unclear whether this DTX3L/TIRR interaction is dose-dependent or time-dependent in response to DNA damage.

Response: We thank the Reviewer for raising these excellent points. Consistent with the XPO1-

TIRR interaction, the interaction between TIRR and DTX3L was also enhanced with increasing IR doses and progressively intensified over time following IR exposure. This suggests that the DTX3L/TIRR interaction is also both dose-dependent and time-dependent in response to DNA damage (**Supplementary Fig. 3g, h and Lines 161-164 page 7**).

11. It would be beneficial to confirm whether the RING domain of DTX3L is responsible for TIRR ubiquitination.

Response: We appreciate the Reviewer's valuable suggestion. Both the RING and DTC domains of DTX3L are reported to play essential roles in its ubiquitylation activity (*Sci Adv.* 2020 PMID: 32937373; *Nat Immunol.* 2015 PMID: 26479788). To investigate this, we constructed DTX3L truncations with deletions of the RING domain (D1), the DTC domain (D2), and both domains (D3). All three deletions resulted in reduced ubiquitylation levels of TIRR as shown in revised **Supplementary Fig. 3k, l and Line 168-173 page 7**.

12. What is the rationale for using His-Ub in Fig S3I.

Response: We thank the Reviewer for pointing this out. Initially, we used His-Ub to enhance TIRR ubiquitination. We have since repeated the experiment without His-Ub, obtaining consistent results, which are now reflected in the updated **Fig. S3p**.

13. It would be preferable to identify the key ubiquitinated sites of TIRR using mass spectrometry analysis. Moreover, conducting additional experiments to confirm that DTX3L cannot degrade the TIRR K187A protein may provide further insights.

Response: We thank the Reviewer for these helpful suggestions. As suggested, we have submitted multiple mass spectrometry samples; however, detecting TIRR ubiquitination has been challenging. Nonetheless, mass spectrometry revealed that TIRR is ubiquitinated at lysine-187, as shown in the revised **Supplementary Fig. 4a**. To further investigate this, we mutated each lysine individually and confirmed that DTX3L promotes polyubiquitination of TIRR at lysine-187 (**Fig. 5a, b**). As suggested, we also demonstrated that both K187R and K187A mutation abolished the DTX3L-mediated TIRR degradation and prolonged the TIRR protein half-life in PC-3 cells (**Fig. 5c, d and Supplementary Fig. 4b-d**).

14. For the NHEJ and HR reporter assays, it would be beneficial to show the efficiency of TIRR knockdown and plasmid transfection. These controls are crucial for quantifying the results of NHEJ and HR reporter assay.

Response: We thank the reviewer for pointing this out. We have examined TIRR protein levels

in **Fig. 5h** to show the efficiency of TIRR knockout and plasmid transfection.

15. There is a concern regarding the inconsistency between the protein levels and the foci of 53BP1 and BRCA1 in Figure S5c~5f.

Response: We thank the Reviewer for raising this concern. The key function of 53BP1 and BRCA1 during DNA damage is their accumulation at damage sites, where they form foci to recruit associated proteins and facilitate NHEJ and HR repair (*Mol Cell*. 2016 PMID: 27546791; *Sci Adv*. 2021 PMID: 34144977; *Nat Commun*. 2022 PMID: 35042897; *Cell Death Differ*. 2019 PMID: 30006610). Consistent with these studies, we did not detect significant changes in their overall protein levels under IR conditions, but instead focused on their foci formation as indicators of DNA repair. To clarify this, we have added relevant statements in the revised manuscript **Lines 238-241 page 10**.

16. DTX3L expression sensitizes prostate cancer cells to synthetic lethality by PARP inhibitors, However, there is no direct evidence linking the function of DTX3L to TIRR.

Response: We thank the Reviewer for raising these excellent points. To address this, we transfected DTX3L-overexpressed cells with TIRR-K187R mutant. We found that while DTX3L overexpression increased cell sensitivity to PARP inhibitors, the TIRR-K187R mutant reduced this sensitivity (**Supplementary Fig. 6a-h and Lines 264-267 page 10**). This suggests that the effect of DTX3L on cell sensitivity to PARP inhibitors depends on its ability to ubiquitinate TIRR.

Minor issues

1. The authors should indicate the molecular weight of proteins for western blots.

Response: We thank the Reviewer for mentioning this point. We have now added all the protein markers in the revised manuscript.

2. The authors should provide more detailed descriptions in the figure legends. This would enhance readability and, consequently, improve the impact and reproducibility of the manuscript.

Response: We thank the Reviewer for pointing this out. We have now added more detailed descriptions in the revised manuscript: **Lines 728, 731, 734, 736, 738, 740, 741 page 27, Lines 747, 748, 751, 755, 757, 761, 763 page 28, Lines 774, 779, 781, 782, 788 page 30, Lines 789 page 31, Lines 796, 800, 801, 804, 805, 806, 808 page 32, Lines 811, 814 page33, Lines 820,, 822, 826, 827, 831, 833 page 34**. So as shown in the revised supplementary materials: **Lines 33, 35 page 2, Lines 46, 53, 55 page 3, Line 62 page 4, Lines 64, 67, 69, 74, 86, 89, 91 page 5**,

Lines 107, 109 page 7 and Line 131 page 8.

Reviewer #2 (Remarks to the Author):

In this manuscript, Ma et al. report their investigation into the regulation of TIRR, a protein that inhibits the DNA damage response protein 53BP1 and plays a key role in facilitating DNA double-strand break repair by homologous recombination. They demonstrate that, following DNA damage, TIRR is modified at lysine 187 (K187) by the E3 ubiquitin ligase DTX3L and exported from the nucleus to the cytoplasm through physical interaction with exportin-1 (XPO1). The nuclear export signal (NES) of TIRR was identified (motif M3 in the manuscript), with K187 adjacent to this NES. Polyubiquitinated TIRR undergoes degradation in the cytoplasm. The authors also show that overexpression of DTX3L in prostate cancer reduces TIRR levels, correlating with inhibition of homologous recombination and chromosomal instability. Reduced TIRR levels sensitize prostate cancer cells to synthetic lethality induced by PARP inhibitors, suggesting that DTX3L overexpression could be a marker for increased sensitivity to PARP inhibitors in prostate cancer. The manuscript is clearly written, well-organized with informative figures, and technically sound. Notwithstanding discrepancies with a preprint in bioRxiv that require attention (see below), the study by Ma et al. will be of great interest to researchers in the DNA repair field.

Response: We thank the Reviewer for recognizing the significance of our finding and the contribution to the new knowledge in the field.

Issue requiring attention

A preprint posted in bioRxiv also reports a detailed characterization of the interaction between TIRR and XPO1 and investigates the TIRR export mechanism (<https://www.biorxiv.org/content/10.1101/2024.02.19.580988v1>). In this preprint, two nuclear export signals, not analyzed by Ma et al., were shown to mediate the interaction of TIRR with XPO1. Furthermore, the preprint indicates that mutations in the NES motif identified by Ma et al. (M3) do not affect the interaction of TIRR with XPO1. Consequently, the authors of the preprint excluded this specific site as an interaction motif for XPO1. These discrepancies are intriguing and should be addressed by Ma et al.

Response: It is noteworthy that this preprint on bioRxiv also investigated XPO1-mediated TIRR nuclear export. We appreciate the Reviewer for highlighting these excellent points. We have addressed these concerns as outlined below.

- a) We noticed that the preprint paper used computer-based websites, including NESmapper and LocNES to predict nuclear export signals (NES) in TIRR. To better predict the

interaction between TIRR and XPO1, we employed AlphaFold 3 (*Nature*. 2024 PMID: 38718835) to assess the XPO1 binding ability of both three NES from the preprint paper (NES1-3) and NES4 from us. As shown in **Fig. R1a**, the NES4 exhibits the strongest affinity for the XPO1 protein with lowest Gibbs free energy (ΔiG) value. The predicted interface of NES4 and XPO1 was shown in **Fig. R1b**.

- b) To experimentally address this, we synthesized both four NES peptides and using Isothermal Titration Calorimetry (ITC) assay further confirmed the binding between NES4 and XPO1, as shown in **Fig. R1c**. Moreover, deletion of NES4 but not other NES abolished the binding of TIRR to XPO1 (**Fig. R1d**).

Fig. R1

Fig. R1 Binding affinity between XPO1 and TIRR nuclear export signals. a Diagram illustrating the nuclear export signal (NES) sequences of TIRR, along with predicted template modeling scores (pTM), surface interface area, and binding free energy (ΔiG) to the XPO1 protein, as determined by AlphaFold 3. **b** Schematic representation highlighting predicted interface between NES4 and XPO1 using AlphaFold 3. **c** Binding affinity measured by ITC between XPO1 and synthesized NES peptides. **d** Co-IP of whole-cell lysates from 293T cells transfected with HA-XPO1, Flag-TIRR-WT or NES mutants.

Reviewer #3 (Remarks to the Author):

I co-review this manuscript with one of the reviewers who provided the listed reports. This is part of nature communications initiative to facilitate the training in peer review and to provide appropriate recognition for Early Career Researcher who co-review manuscripts.

Response: We very much appreciate the reviewer for the positivity and enthusiasm about our

study.

Reviewer #4 (Remarks to the Author):

The balance between homologous recombination (HR) and non-homologous end-joining (NHEJ) for repairing DNA double-stranded breaks (DSBs) has been well-established, and serves as one of the main regulatory mechanisms is through 53BP1 recruitment to the DSB sites and suppressing BRCA1-mediated HR. In this manuscript, Ma et al. uncovered an additional layer of 53BP1 regulation, which is through the subcellular localization changes in the 53BP1-interacting protein TIRR. The nuclear export of TIRR is triggered by ubiquitination of its TES domain in response to DNA damage, and this modification is required for XPO1 interaction to mediate TIRR export. Moreover, the authors identified the E3 ubiquitin ligase responsive to the ubiquitination as well as a ubiquitination site that plays a major role in the pathway. The data provided generally support their proposed model after some points are addressed. In particular, the exact subcellular location where TIRR degradation happens needs clarification, as detailed further below. Finally, the results from patient samples and animal models suggest the potential of targeting this pathway for PARP inhibitor sensitization. Although the overall novelty and the significance of this study are moderate, the well-prepared manuscript, and the robust, complete, and clear presentation of experimental data mostly compensate for this modest weakness.

Response: We thank the Reviewer for the positive comments on our manuscript and great suggestions.

Below are the major points for this study:

1. It's unclear whether the TIRR protein degradation happens in the nucleus or cytoplasm. Although the author proposed the degradation happens after translocation to cytosol, the evidence is insufficient. For example, did they see less protein degradation after DNA damage when knocking out XPO1? Is DTX3L only expressed in the nucleus? Additionally, inhibiting ubiquitination by MLN4924 not only induces nuclear accumulation but also increases the total protein level of TIRR (Figure 3e-g). Does this suggest that some degradation can also happen in the nucleus? The proteasome is known to be both cytoplasmic and nuclear, although perhaps mostly the former.

Response: We thank the Reviewer for raising these excellent points. As shown in Fig. 4m, DTX3L is expressed in both nucleus and cytoplasm, indicating that its localization may not be the primary factor driving TIRR degradation in the cytoplasm. As suggested, we found that XPO1 knockout prevented TIRR degradation following DNA damage (Fig. 3j), further suggesting that TIRR degradation likely occurs in the cytoplasm.

2. Regarding TIRR ubiquitination, the data presented in Figure 3 are mostly done by exogenous proteins. Did they confirm if endogenous TIRR can also be ubiquitinated by DTX3L with endogenous ubiquitin pulldowns (i.e., a TUBE or Ub pulldown to complement Figure 4i)? This would strengthen the manuscript. Also, did they confirm the ubiquitination level of TIRR after treating with MLN4924 and MG132?

Response: We appreciate the Reviewer's valuable suggestion. As recommended, we performed an endogenous tandem ubiquitin-binding entity (TUBE) pulldown assay, which confirmed that TIRR undergoes ubiquitination in response to DNA damage (**Fig. 4i**), particularly after treatment with MG132, but not with MLN4924 (**Supplementary Fig. 3i**). Additionally, DTX3L knockout reduced endogenous TIRR ubiquitination, as shown in the revised **Fig. 4i**.

3. It would improve the manuscript if they rescued their DTX3L knockout with WT and a catalytically inactive mutant to further prove that the E3 ligase activity of DTX3L is ubiquitinating TIRR.

Response: We thank the Reviewer for these helpful suggestions. As reviewer suggested, we constructed several previously reported catalytically inactive mutant of DTX3L (*Nat Immunol.* 2015 PMID: 26479788) and found that C561S/C564S mutant of DTX3L completely abolishes TIRR ubiquitination, as shown in the revised **Supplementary Fig. 3m**.

4. Figure 4e, the author should confirm whether the endogenous TIRR has a similar response to DTX3L overexpression and MG132 treatment, as compared to the exogenous TIRR.

Response: We thank the Reviewer for raising these excellent points. As suggested, we repeated this experiment using TIRR antibody to detect both endogenous and exogenous TIRR level. The endogenous TIRR level exhibited a consistent change with the exogenous TIRR, though it was less pronounced, as shown in the revised **Fig. 4e**.

5. The K187R mutant seems to have a dominant effect on the 53BP1-mediated DSB repair pathway choices. It would be interesting if at least one of the other mutants used in Figure 5b would behave more like wildtype TIRR, to further confirm the specificity of K187 ubiquitination.

Response: We appreciate the Reviewer's valuable suggestion. As recommended, we further investigated two additional TIRR mutations (K170R, K198R) and demonstrated that they have a similar effect on 53BP1 foci inhibition as TIRR-WT, as shown in the revised **Fig. 5h-l**.

6. Figure 6i-j, why does DTX3L expression induce more chromosome breaks? The authors need to further explain why this phenotype is associated with the shift of two DSB repair pathways. If not, what could be the possible reason?

Response: We thank the Reviewer for pointing this out. The increased frequency of asymmetric radial chromosome structures observed in HR-defective cells is indicative of NHEJ-mediated chromosome exchange, a hallmark of HR deficiency. This phenotype's association with shifts between the HR and NHEJ pathways has been demonstrated by our group and others through studies in both cell and mouse models. (*Cell*. 2018 PMID: 29656893; *Cell*. 2010 PMID: 20362325; *Sci Adv*. 2021 PMID: 34144977; *Nat Commun*. 2023 PMID: 37002234). As suggested, we have added a description and reference in the revised manuscript **Lines 243-246 page 10**.

Minor points:

7. In Figure 1c, why does the overall intensity of 53BP1 increase after DNA damage? The point that the authors want to make is that 53BP1 “foci” are increased upon DNA damage. However, the representative images did not show clear foci formation and the data looks more like just the overall intensity elevation. The authors need to clearly describe how they define the foci positive cells, and how they distinguish this from an overall intensity change. Also, the corresponding figure legend needs to be fixed, it’s a co-staining of Flag and “53BP1”, not with p-H2AX. How to distinguish the DSB induced foci from an overall intensity change in the cells upon irradiation, and how to define foci positive cells?

Response: We appreciate the Reviewer for highlighting these excellent points. We have addressed these concerns as outlined below.

- a) Protein foci, such as 53BP1 or γ -H2AX, form as a result of protein accumulation in response to DNA damage. This accumulation results in a stronger IFC signal in the IR group, whereas the group without IR exhibits a weaker signal due to the protein being uniformly distributed (*Nature*. 2023 PMID: 37853125). To prevent any misinterpretation, we replaced the image with a relatively stronger one, as shown in the revised **Fig. 1c**.
- b) To differentiate DSB-induced foci from overall intensity changes, we analyzed IFC images and quantified the foci using ImageJ. As shown in **Fig. R2** below, the uniformly distributed signal was removed to more accurately identify and count the foci. This allowed for a clearer distinction of the overall intensity at this stage.

Fig. R2

Fig. R2 A schematic diagram illustrating how to use ImageJ for processing immunofluorescence microscopy data of double-strand break (DSB)-induced foci. **a** Original images showing 53BP1 in U2OS cells were captured after treatment with or without IR. **b** Set a threshold and use the 'Analyze Particles' function to segment the cells to analyze 53BP1 signal intensity. **c** Uniformly distributed signals were removed based on the threshold value. **d** The 'Find Maxima' function was used to quantify 53BP1 foci signals. Cells treated with IR exhibited an increase in 53BP1 foci signals.

c) We apologize for the mistake in the figure legend. It has been corrected as indicated in **Line 731 page 27** in the revised manuscript.

8. Figure 6e: it seems like the overall intensity of BRCA1 is decreased by DTX3L. Again, the author needs to clarify the changes they observed are reflected by foci number, not the overall intensity.

Response: We thank the Reviewer for pointing this out. As mentioned above, we replaced the image with a relatively stronger one, as shown in the revised **Fig. 6e**.

Reviewer #5 (Remarks to the Author):

Response: We thank the Reviewer for the positive comments on our manuscript and great suggestions.

Authors' Response to Reviewers' Comments on Manuscript NCOMMS-24-19963B

We thank the Editor and the Reviewers for the improvement of our manuscript.

REVIEWERS' COMMENTS

Reviewer #1 (Remarks to the Author):

My comments have been addressed in the revised version. I support the publication of this manuscript.

Response: We thank the reviewer for your insightful suggestions, which have greatly enhanced our manuscript.

Reviewer #2 (Remarks to the Author):

The authors have adequately addressed my comments. Using quantitative binding assays (ITC) with TIRR peptides, they clearly demonstrate that only NES4 interacts with XPO1, whilst the other putative binding sequences (NES1, NES2, and NES3 peptides) show no affinity for XPO1. Co-IP of whole-cell lysates with HA-XPO1 and Flag-TIRR also support these findings.

The K_d value for NES4 peptide should be included on the ITC graph or in the figure legend. Additionally, it would make sense to incorporate these critical ITC and co-IP data into one of the figures in the manuscript.

Overall, this is a well-executed study that is suitable for publication in Nature Communications.

Response: We thank the Reviewer for the positive comments on our manuscript. As suggested, we have included the K_d value for NES4 peptide on the ITC graph and included the data in the revised Supplementary Fig. 2c.

Reviewer #3 (Remarks to the Author):

Response: We thank the Reviewer for the improvement of our manuscript.

Reviewer #4 (Remarks to the Author):

The authors have done a thorough job with addressing all of the previous concerns. I believe the manuscript should now be accepted.

Response: We thank the Reviewer for the improvement of our manuscript.

Reviewer #5 (Remarks to the Author):

Response: We thank the Reviewer for the improvement of our manuscript.